# Robust Spectral Watermark for Synthetic Tabular Data

Yizhou Zhao
University of Pennsylvania

Xiang Li
University of Pennsylvania

Peter X. K. Song
University of Michigan

Qi Long
University of Pennsylvania

Weijie Su*
University of Pennsylvania

**Abstract**

The rise of generative AI has enabled the production of high-fidelity synthetic tabular data across fields such as healthcare, finance, and public policy, raising growing concerns about data provenance and misuse. Watermarking offers a promising solution to address these concerns by ensuring the traceability of synthetic data, but existing methods face many limitations: they are computationally expensive due to reliance on the inverse process of large diffusion models, struggle with mixed discrete-continuous data, or lack robustness to common post-processing attacks. To address these limitations, we propose Tab-Drw, an efficient and robust post-editing watermarking scheme for synthetic tabular data. Tab-Drw embeds watermark signals in the frequency domain: it normalizes heterogeneous features via the Yeo-Johnson transformation and standardization, applies the discrete Fourier transform (DFT), and adjusts the imaginary parts of adaptively selected entries according to precomputed pseudorandom bits. To further enhance robustness and efficiency, we introduce a novel rank-based pseudorandom bit generation method that enables row-wise retrieval without incurring storage overhead. Experiments on five benchmark tabular datasets show that Tab-Drw achieves strong detectability and robustness against post-processing and adaptive attacks, while preserving high data fidelity and fully supporting mixed-type features.

**Keywords:** Watermarking, Synthetic Tabular Data, Robustness, Generative AI

**Mathematics Subject Classification (2020):** Primary 62H15; Secondary 62F35, 62H12.

## 1   Introduction

Tabular data is a predominant format for structured information in many fields such as healthcare, finance, and public policy (Borisov et al., 2022). It facilitates tasks such as decision-making, risk assessment, and resource allocation. However, access to high-quality tabular data is often restricted by privacy concerns, regulatory constraints on data sharing, and the cost of human annotation. Recent advances in generative AI have revolutionized synthetic data generation (Xu et al., 2019; Zhao et al., 2021; Zhang et al., 2021; Liu et al., 2023; Kotelnikov et al., 2023; Zhang

---

*Corresponding author: suw@wharton.upenn.edu

et al., 2024a; Gulati and Roysdon, 2024; Wang and Nguyen, 2025), creating high-fidelity tabular datasets that closely match real-world data. Synthetic tabular data now offers a compelling alternative for data sharing and model training in various domains (Bauer et al., 2024).

Despite these benefits, synthetic tabular data introduces additional risks. Misuse may lead to civil disputes, regulatory violations, or broader societal harms (Guo and Chen, 2024). For example, generating synthetic datasets from copyrighted sources without authorization may infringe intellectual property rights (Vyas et al., 2023). In finance, synthetic transaction records can be used to facilitate fraud (Assefa et al., 2020). In healthcare, biased or inaccurate synthetic patient data may misguide clinical decision-making and result in adverse outcomes (Qian et al., 2024). As synthetic data becomes increasingly realistic and widespread, ensuring accountability and provenance has become critical (Liu et al., 2024).

To address concerns surrounding the misuse of synthetic tabular data, watermarking has emerged as a promising solution. The core idea is to embed invisible statistical signals into synthetic data before release, allowing reliable detection by a verifier with access to secretly shared information. An effective watermarking scheme should satisfy four key properties: 1) **fidelity**, preserving the quality and utility of the data, 2) **detectability**, allowing reliable identification through a private detection process, 3) **applicability**, enabling efficient watermark embedding even after data generation and for mixed discrete-continuous tabular data, and 4) **robustness**, ensuring resilience against post-processing attacks such as deletions or value modifications (Podilchuk and Delp, 2001; Kuditipudi et al., 2024; Li et al., 2025b). Although significant progress has been made in watermarking text data (Aaronson, 2023; Kirchenbauer et al., 2023; Kuditipudi et al., 2024; Zhao et al., 2024; Dathathri et al., 2024; Block et al., 2025; Zhao et al., 2025) and images (Wen et al., 2023; Yang et al., 2024; Zhang et al., 2024b), existing watermarking schemes for synthetic tabular data fail to simultaneously achieve these four properties.

## 1.1 Existing Work

Current approaches to watermarking synthetic tabular data mainly fall into two categories: sampling-phase watermarking (Zhu et al., 2025; Fang et al., 2025) and post-editing watermarking (He et al., 2024; Zheng et al., 2024). Sampling-phase methods typically change the sampling process of large diffusion models. Specifically, Zhu et al. (2025) embeds watermark signals into structured latent noise and detects the structure by measuring correlations with noise reconstructed via the inverse process. While achieving high fidelity and robustness, it relies on reversible sampling strategies, such as DDIM (Song et al., 2021), which is prone to reconstruction errors and computationally expensive. Fang et al. (2025) generates multiple samples at the same time and outputs the one with the highest pseudorandom score, which incurs higher computational cost though preserving generation quality.

In contrast, post-editing methods are lightweight: they often modify generated or existing datasets with pseudorandom operations, but don't change the sampling process or invoke large neural networks. For example, He et al. (2024) proposes a "green list" method that bins each tabular value into key-selected intervals and detects whether values fall into the designated sets. Although effective in preserving fidelity and detectability, it struggles with mixed-type

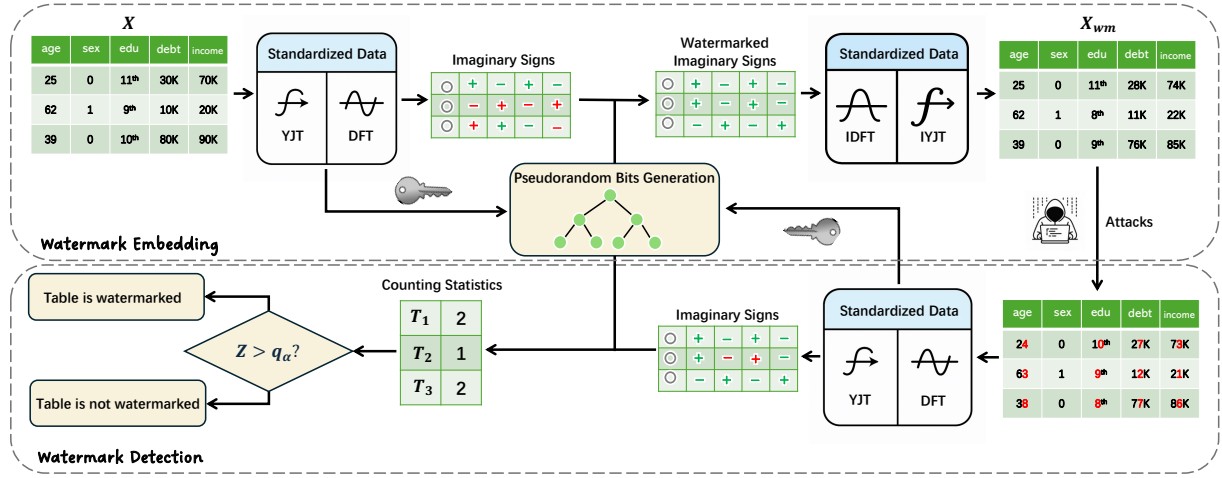

Figure 1: Our proposed watermarking scheme, TAB-DRW, embeds watermarks into standardized tabular data by modifying the imaginary components of the frequency-domain representation to align with pseudorandom bits. Detection evaluates the degree of alignment: strong alignment indicates watermarked data, while weak alignment suggests non-watermarked data.

Table 1: Comparison of our method with existing works. ○ indicates "not satisfied", ◐ indicates "partially satisfied", and ● indicates "satisfied".

| Methods | Category | Data type | Fidelity | Detectability | Applicability | Robustness |
|---|---|---|---|---|---|---|
| Zhu et al. (2025) | Sampling-phase | Continuous & Discrete | ● | ● | ◐ | ◐ |
| Fang et al. (2025) | Sampling-phase | Continuous & Discrete | ● | ● | ◐ | ◐ |
| He et al. (2024) | Post-editing | Only continuous | ● | ● | ◐ | ◐ |
| Zheng et al. (2024) | Post-editing | Continuous & Discrete | ● | ● | ◐ | ◐ |
| **TAB-DRW (Ours)** | **Post-editing** | **Continuous & Discrete** | ● | ● | ● | ● |

(continuous and discrete) data and lacks robustness against noise attacks. Zheng et al. (2024) embeds watermark signals by perturbing key cells with values randomly selected from the so-called "green domain". However, it requires storing the original dataset for perturbation recovery, leading to substantial space overhead in generative settings.

Overall, existing methods offer valuable insights but fall short of providing a lightweight, robust, and broadly applicable solution for synthetic tabular data. See Appendix A for a broader discussion of related work on watermarking.

## 1.2 Our Contribution

In this work, we propose a new watermarking method that simultaneously satisfies the four desired properties above. Our contributions are summarized below.

1. **A new watermarking scheme.** We propose TAB-DRW, a lightweight post-editing watermarking method that embeds robust watermark signals in the frequency domain. Specifically, it modifies the discrete Fourier transform (DFT) representation of tabular data to align with a precomputed pseudorandom bit sequence (see the first row of Figure 1, detailed in Section 2). Detection then evaluates the degree of alignment: strong alignment indicates watermarked data, while weak alignment suggests non-watermarked data (see the second row of Figure 1, detailed in Section 2).

Tab-Drw offers several practical advantages. First, it is computationally efficient and requires no access to the data-generating model. Second, it applies to both existing and synthetic tabular data while preserving data fidelity with minimal distortion. Third, it is versatile and can be readily extended to other settings. Optional rounding and outlier clipping are applied to maintain semantic validity. To support real-world multi-key deployment and mitigate informative attacks, we further introduce a privacy-enhanced variant.

2. **A new pseudorandom bit generation method.** In Tab-Drw, we propose a rank-based pseudorandom bit generation scheme for embedding a secret yet detectable signal. The key idea is to leverage rank information to enhance robustness against post-processing attacks, while avoiding explicit storage of the pseudorandom bits (see Figures 2 and 3, and Section 2 for details). Under moderate attacks, the scheme guarantees that the recovered pseudorandom bits change only minimally. The method is also computationally efficient: during detection, the pseudorandom bits are reconstructed by querying an implicit storage structure induced by robust statistics computed on key-selected columns.

3. **Theoretical analysis.** We provide theoretical insights into the bias and robustness of Tab-Drw. We characterize the bias through closed-form expressions for the resulting entry-wise and column-wise distortion, and establish robustness by deriving a lower bound on the Z-score under noise-corrupted watermarked tabular data in realistic settings.

4. **Empirical validation and robustness evaluation.** We show that Tab-Drw consistently achieves superior detectability and robustness while preserving high data fidelity compared to existing methods. This is validated through comprehensive experiments on five benchmark datasets with mixed feature types. We further evaluate robustness against informed adversaries who know the full watermarking pipeline but not the secret key, and demonstrate that targeted watermark scrubbing or spoofing cannot be performed reliably without incurring substantial loss of data fidelity. Finally, through a case study on a real-world dataset, we show that Tab-Drw is entropy-aware, with embedding strength implicitly adapting to feature entropy to preserve semantics for low-entropy features (e.g., `gender`).

### 1.3 Organization of the Paper

The remainder of the paper is organized as follows. In Section 2, we introduce the proposed Tab-Drw, including the frequency-domain watermark embedding and detection procedures, and the rank-based pseudorandom bit generation scheme. In Section 3, we provide a theoretical analysis of watermark distortion and robustness under additive Gaussian noise. In Section 4, we present empirical evaluations on five benchmark datasets, covering fidelity, detectability, robustness to post-processing and adaptive attacks, watermark embedding and detection runtime, and a case study on the impact of watermark embedding on low-entropy categorical variables. Finally, we conclude in Section 5 with a discussion and directions for future research. Most experimental implementation details, supplementary results, and technical proofs are provided in the appendix.

## 2 Method

### 2.1 Watermark Embedding

**High-level description.** As shown in Figure 1, the embedding process can be divided into three steps. It begins by preprocessing the given tabular data through two transformations: first, a column-wise Yeo-Johnson transformation (YJT) (Yeo and Johnson, 2000) with standardization to reduce heterogeneity and unify the scale; second, a row-wise discrete Fourier transform (DFT) (Oppenheim, 1999) to map the data into the frequency domain. Next, the algorithm modifies the imaginary components of the frequency-domain representation to align with a precomputed pseudorandom bit sequence. Finally, it applies the inverse transformations to reconstruct the modified data in the original domain, which is then released for public use and future detection. We provide a detailed explanation of these steps below. [1]

**Definition 1** (YJT). *For a real-valued input $x \in \mathbb{R}$, the YJT $\Psi(\lambda, x)$ is defined as*

$$\Psi(\lambda, x) = \begin{cases} \frac{(x+1)^{\lambda}-1}{\lambda}, & \text{if } x \geq 0, \ \lambda \neq 0, \\ \ln(x+1), & \text{if } x \geq 0, \ \lambda = 0, \\ -\frac{(-x+1)^{2-\lambda}-1}{2-\lambda}, & \text{if } x < 0, \ \lambda \neq 2, \\ -\ln(-x+1), & \text{if } x < 0, \ \lambda = 2. \end{cases}$$

*Here, $\lambda$ is a transformation parameter that is automatically selected to reduce heterogeneity (SciPy Developers, 2025).*

**Definition 2** (DFT and IDFT). *Given a row of tabular data $\boldsymbol{x} = (x_0, x_1, \ldots, x_{p-1}) \in \mathbb{R}^p$, its DFT is defined as $\boldsymbol{y} = \text{DFT}(\boldsymbol{x}) := (y_0, y_1, \ldots, y_{p-1}) \in \mathbb{C}^p$, where for each $t = 0, \ldots, p-1$,*

$$y_t = \frac{1}{\sqrt{p}} \sum_{n=0}^{p-1} x_n e^{-\mathrm{i}\frac{2\pi}{p}tn}.$$

*The inverse DFT (IDFT) is given by $\boldsymbol{x} = \text{IDFT}(\boldsymbol{y}) \in \mathbb{R}^p$, where for each $n = 0, \ldots, p-1$,*

$$x_n = \frac{1}{\sqrt{p}} \sum_{k=0}^{p-1} y_k e^{\mathrm{i}\frac{2\pi}{p}kn}$$

*Since $\text{IDFT} \circ \text{DFT} = \text{Id}$, their composition implies an exact recovery of the original input.*

**Step 1: Column-wise and row-wise transformations.** In general, features in a tabular dataset exhibit heterogeneous scales and types (continuous or discrete). This heterogeneity would prevent a uniform watermarking process among features, as features with larger magnitudes could dominate others. To address this, we first apply a column-wise YJT defined in Def. 1, and

---

[1]Throughout this paper, we denote the imaginary unit by i, to avoid confusion with the index $i$.

then standardize each transformed column. A crucial property of YJT is that it is monotonic and invertible, allowing for exact recovery of the original data through its inverse. After the YJT followed by standardization, each row becomes a real-valued sequence with unified scale. We then apply a row-wise DFT to obtain the frequency-domain representation $\boldsymbol{y} = \mathtt{DFT}(\boldsymbol{x}) \in \mathbb{C}^p$, as defined in Def. 2.

**Step 2: Modification on the imaginary parts of the DFT.** Let $\boldsymbol{x} := (x_0, x_1, \ldots, x_{p-1})$ denote a row of tabular data $\mathbf{X} \in \mathbb{R}^{N \times p}$ after applying the YJT and standardization. Let $\boldsymbol{y} := \mathtt{DFT}(\boldsymbol{x}) = (y_0, y_1, \ldots, y_{p-1}) \in \mathbb{C}^p$ be the frequency-domain representation of $\boldsymbol{x}$ obtained via the DFT. Since $\boldsymbol{x}$ is real-valued, its DFT coefficients satisfy conjugate symmetry, that is, $y_t = \overline{y_{p-t}}$ for $t = 1, \ldots, p-1$. As a result, the coefficients with indices $t$ and $p-t$ form conjugate pairs. The coefficient $y_0$, corresponding to the DC component, is real-valued and therefore does not form a pair with any other index. In addition, when $p$ is even, the coefficient $y_{p/2}$ corresponds to the Nyquist frequency and is also real-valued, making it self-conjugate. Consequently, it suffices to modify only the entries of $\boldsymbol{y}$ with indices $\{1, \ldots, m\}$, where $m = \left\lfloor \frac{p-1}{2} \right\rfloor$, as all remaining coefficients are uniquely determined by conjugate symmetry. We refer to the entries $\{y_t\}_{t=1}^m$ as the effective entries. For each entry $y_t \in \mathbb{C}$, we denote its real and imaginary parts by $\Re(y_t)$ and $\Im(y_t)$ respectively. For each effective entry, we generate a 0-1 pseudorandom bit $\zeta_t \sim \mathrm{Bernoulli}(0.5)$ and modify $\Im(y_t)$ to align with the corresponding $\zeta_t$. We consider two modification strategies, described below.

**Initial idea: Hard sign flip.** The most natural strategy is to force the sign of $\Im(y_t)$ to match $\zeta_t$. Specifically, for each $t = 1, \ldots, m$, we define:

$$y_t^{\mathrm{wm}} = \Re(y_t) + (2\zeta_t - 1)\mathrm{i} \cdot |\Im(y_t)|, \quad \text{and} \quad y_{p-t}^{\mathrm{wm}} = \overline{y_t^{\mathrm{wm}}}. \tag{1}$$

Under this rule, if $\Im(y_t)$ already matches the sign of $\zeta_t$, no change is made and $y_t^{\mathrm{wm}} = y_t$. Otherwise, the sign of $\Im(y_t)$ is flipped so that $\Im(y_t^{\mathrm{wm}}) = -\Im(y_t)$.

**Refinement: Soft variant.** The hard sign flipping may potentially introduce large distortions, degrading data fidelity. As a refinement, we introduce a softer modification controlled by two soft hyperparameters $(\gamma, \delta)$. Specifically, we modify $y_t$ only if $|\Im(y_t)|$ is among the $\gamma$-smallest values in $\{|\Im(y_t)|\}_{t=1}^m$ and the sign of $\Im(y_t)$ differs from $2\zeta_t - 1$. Furthermore, we shrink the imaginary part by a factor $\delta \in [-1, 1]$ to further limit the distortion:

$$y_t^{\mathrm{wm}} = \begin{cases} \Re(y_t) - \mathrm{i}\delta \cdot \Im(y_t), & \text{if } \Im(y_t) \cdot (2\zeta_t - 1) < 0 \text{ and } |\Im(y_t)| \leq \mathrm{Quantile}_\gamma(\{|\Im(y_t)|\}_{t=1}^m), \\ y_t, & \text{otherwise,} \end{cases} \tag{2}$$

and $y_{p-t}^{\mathrm{wm}} = \overline{y_t^{\mathrm{wm}}}$. When $\gamma = \delta = 1$, the soft variant in (2) reduces to the hard sign flip in (1). When $\gamma = 0$ or $\delta = -1$, it reduces to no watermarking. A more fine-grained theoretical analysis of how $(\gamma, \delta)$ influences the resulting watermark distortion is provided in Section 3. In practice, varying $(\gamma, \delta)$ enables flexible control over the trade-off between watermark strength and data fidelity. For example, one can tune $(\gamma, \delta)$ to maximize detectability under a prescribed

---

**Algorithm 1** Watermark embedding of TAB-DRW

---

1: **Input**: Tabular data $\mathbf{X} \in \mathbb{R}^{N \times p}$, parameters $\gamma \in [0, 1]$ and $\delta \in [-1, 1]$.
2: **Initial**: Transform $\mathbf{X}$ using YJT and standardization (still denoted as $\mathbf{X}$ for simplicity).
3: **for** each row $\boldsymbol{x}$ in $\mathbf{X}$ **do**
4:      Compute $\boldsymbol{y} \leftarrow \mathtt{DFT}(\boldsymbol{x})$ and generate pseudorandom bits $\{\zeta_t\}_{t=1}^m$ via Algorithm 2.
5:      Modify $\boldsymbol{y}$ according to soft variant (2) to obtain $\boldsymbol{y}^{\mathrm{wm}}$.
6:      Recover $\boldsymbol{x}^{\mathrm{wm}} \leftarrow \mathtt{IDFT}(\boldsymbol{y}^{\mathrm{wm}})$.
7: **end for**
8: Collect each $\boldsymbol{x}^{\mathrm{wm}}$ to form a matrix $\mathbf{X}^{\mathrm{wm}}$.
9: Apply inverse standardization and inverse YJT to $\mathbf{X}^{\mathrm{wm}}$, round and clip if needed, and release.

---

distortion budget by performing a lightweight grid search on synthetic samples. Detailed runtime evaluations are reported in Section 4.5.

**Step 3: Inverse steps to return to the original data domain.** In the final step, we apply the inverse DFT to each modified $\boldsymbol{y}^{\mathrm{wm}}$. Since $\boldsymbol{y}^{\mathrm{wm}}$ preserves conjugate symmetry, the inverse DFT yields a real-valued vector. We then collect the resulting vectors to form a matrix $\mathbf{X}^{\mathrm{wm}}$ and apply the inverse standardization followed by the inverse YJT to map the data back to the original domain. For discrete features, we round values to the nearest valid entry. For example, a value of 0.4 for the `gender` entry would be rounded to 0 (`Female`), while 0.6 would be rounded to 1 (`Male`). For bounded features, we clip values to stay within the valid range. The full procedure is summarized in Algorithm 1.[2] In Appendix G.1, we present an ablation study examining the impact of necessary rounding and clipping on watermark detectability. In Section 4.6, we provide a case study to illustrate how TAB-DRW handles low-entropy categorical variables, such as `gender`, in a conservative and adaptive manner that preserves their semantic validity.

*Remark* 1 (Related work). Modifying the frequency-domain representation to embed watermarks has also been explored for diffusion models, such as Tree-Ring Watermarking (Wen et al., 2023). However, this method typically applies deterministic, structured modifications (e.g., zeroing subregions), which are unsuitable for tabular data where each feature has distinct semantic meanings. In contrast, TAB-DRW performs fine-grained, row-wise perturbations guided by pseudorandom bits, preserving feature fidelity while enabling robust detection through a rank-based pseudorandom bit generation scheme.

*Remark* 2 (Column selection for watermarking). Since TAB-DRW is a lightweight post-editing watermark, model providers or dataset owners can flexibly choose any subset of columns to watermark. For example, the watermark can be applied only to columns containing sensitive or high-value information that attackers are unlikely to modify. In Section 4, we do not use any specialized column-selection strategy and exclude only columns with extreme distributions,

---

[2]We also introduce a privacy-enhanced variant to support multi-key scenarios. See Appendix B & G.3 for details.

which contribute little to the watermark signal and can cause scaling issues in a small number of rows even after YJT.

## 2.2 Watermark Detection

During detection, we can recover the pseudorandom bits using secret keys. Recall that TAB-DRW embeds watermark signals by flipping the imaginary signs in the frequency domain via the alignment with these pseudorandom bits. As a result, watermarked rows are expected to exhibit stronger alignment with the recovered pseudorandom bits. A natural detection strategy is to count the number of aligned entries in the suspect data under investigation: if the alignment is significantly higher than the expected without watermarking, we declare the data watermarked; otherwise we do not. Statistically speaking, we solve the following hypothesis testing problem (He et al., 2024):

$$H_0 : \text{ The table is not watermarked} \quad \text{vs.} \quad H_1 : \text{ The table is watermarked.}$$

Given a suspect tabular data $\mathbf{X} \in \mathbb{R}^{N \times p}$, we first apply **Step 1** of the watermark embedding procedure to obtain its frequency-domain representation $\mathbf{Y} = \{y_{i,j}\}_{i,j} \in \mathbb{C}^{N \times p}$ , and denote the corresponding pseudorandom bits by $\{\zeta_{i,j}\}_{i,j}$. For each row $i$, we define the alignment count

$$T_i = \sum_{j=1}^{m} \mathbb{I}\left[\Im(y_{i,j}) \cdot (2\zeta_{i,j} - 1) > 0\right].$$

Then we compute a one-sided Z-score to measure deviation from the expected alignment under $H_0$:

$$Z = \frac{\frac{1}{N}\sum_{i=1}^{N} T_i - \mu_{\mathrm{nwm}}}{\frac{\sigma_{\mathrm{nwm}}}{\sqrt{N}}}, \tag{3}$$

where $\mu_{\mathrm{nwm}}$ and $\sigma_{\mathrm{nwm}}$ denote the mean and standard deviation of $T_i$ under $H_0$. Given a critical value $q_\alpha$ for a significance level $\alpha$, we reject $H_0$ and declare the table watermarked if $Z > q_\alpha$. In practice, we approximate $\mu_{\mathrm{nwm}}, \sigma_{\mathrm{nwm}}$ and $q_\alpha$ using Monte Carlo simulation.

## 2.3 Pseudorandom Bits Generation

The remaining issue is how to construct the pseudorandom bits $\{\zeta_{i,j}\}_{i,j}$ used for watermark embedding and detection. The design must satisfy two key requirements: 1) **robustness**, ensuring that recovered pseudorandom bits remain stable under post-processing attacks, and 2) **memory efficiency**, avoiding the impractical cost of explicitly storing pseudorandom bits for each generated table.

**Scheme description.** To achieve the two goals, we propose a new pseudorandom bit generation scheme with two key components: 1) an implicit storage structure based on a binary tree, and 2) a retrieval mechanism using rank statistics. Algorithm 2 presents the procedure for generating the pseudorandom bit sequence for each row. Below, we describe the scheme step by step.

Given a standardized tabular data $\mathbf{X} \in \mathbb{R}^{N \times p}$ with $m = \lfloor \frac{p-1}{2} \rfloor$ effective entries, we first sample a subset of column indices $\mathcal{I} \subset \{0, 1, \ldots, p-1\}$ using the secret watermark key $\kappa$ (Line 3).

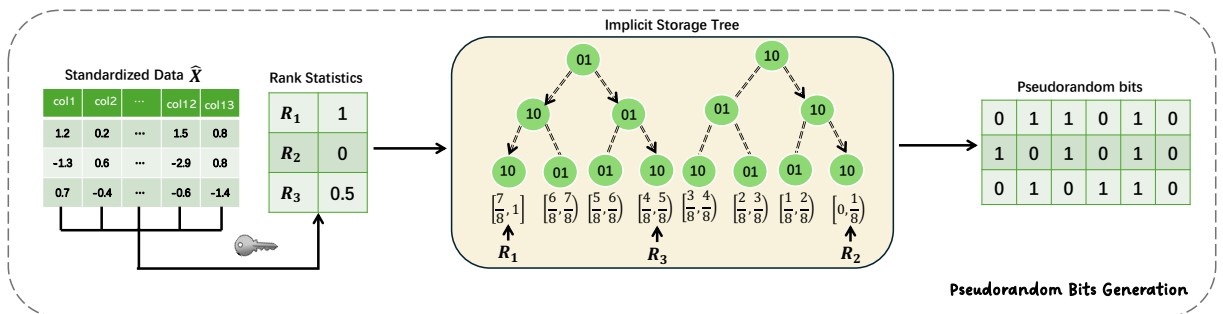

Figure 2: In the proposed pseudorandom bit generation scheme, bit sequence for each row is generated by mapping a normalized row-wise rank statistic to a leaf node in a binary tree.

---

**Algorithm 2** Row-wise Pseudorandom Bits Generation

1: **Input**: Standardized tabular data $\mathbf{X} \in \mathbb{R}^{N \times p}$, target row $\boldsymbol{x}^* \in \mathbb{R}^{1 \times p}$, and watermark key $\kappa$.
2: **Initial**: An empty pseudorandom bit list $\mathbf{S}$, $m = \lfloor (p-1)/2 \rfloor$.
3: Sample a subset of column indices $\mathcal{I} \subset \{0, 1, \ldots, p-1\}$ using $\kappa$.
4: Compute the sum of selected entries for each row of $\mathbf{X}$.
5: Compute the rank of the target row among all rows in $\mathbf{X}$ to obtain $x^*_{\mathrm{rank}}$.
6: Normalize $x^*_{\mathrm{rank}}$ to lie in $[0, 1]$: $x^*_{\mathrm{rank}} \leftarrow x^*_{\mathrm{rank}}/(N-1)$.
7: **for** $j \leftarrow 1$ **to** $\lceil m/2 \rceil$ **do**                    ▷ Traverse the path from the root to the leaf.
8:     Locate the underlying node in the path: $k \leftarrow \min\left(2^j - 1, \lfloor 2^j \cdot x^*_{\mathrm{rank}} \rfloor\right)$.
9:     Append $[1, 0]$ to $\mathbf{S}$ if $k\%4 = 0$ or $3$, else $[0, 1]$.
10: **end for**
11: Truncate $\mathbf{S}$ to its first $m$ entries, and release.

---

For each row $\boldsymbol{x}^* \in \mathbb{R}^{1 \times p}$, we compute the sum of its entries over $\mathcal{I}$ and use this value to determine $x^*_{\mathrm{rank}}$, the rank of the row among all rows in $\mathbf{X}$. The resulting rank is then normalized to the interval $[0, 1]$ (Lines 4–6). For example, if a row has rank $x^*_{\mathrm{rank}} = 1$ among $N = 3$ rows (i.e., the second-largest), its normalized rank is $x^*_{\mathrm{rank}}/(N-1) = 0.5$. Next, we partition $[0, 1]$ into $2^{\lceil m/2 \rceil}$ equal-sized bins and construct a binary tree of depth $\lceil m/2 \rceil$, where each node is deterministically assigned a pseudorandom bit pair and each leaf node corresponds to one bin. The normalized rank of each row determines its assigned bin, and the path from the root to the corresponding leaf encodes the pseudorandom bit sequence for that row (see Figure 2 for an illustration through a simplified example with standardized tabular data $X \in \mathbb{R}^{3 \times 13}$ and $m = 6$). Specifically, at each level $j$ of the tree, we use $x^*_{\mathrm{rank}}$ to identify the associated node and append its bit pair to the sequence list $\mathbf{S}$ (Lines 7–9; see Figure 3 for an illustration). Lines 8–9 jointly specify the node–bit assignment policy and the coupling between the $2^{\lceil m/2 \rceil}$ bins and their corresponding leaf nodes. Finally, we truncate $\mathbf{S}$ to its first $m$ entries to obtain the pseudorandom bit sequence for the target row. A concrete example illustrating each step of the procedure is provided in Appendix C.

**Robustness of the pseudorandom bits.** Our pseudorandom bit generation is robust against perturbations owing to two mechanisms. First, the subset $\mathcal{I}$ of columns used for score computation is determined by the secret key. An adversary without it cannot targetedly modify the specific entries that contribute to pseudorandom bit generation. Second, the sum-based rank statistic is highly stable, so small perturbations to a subset of columns (even those within $\mathcal{I}$) often do not

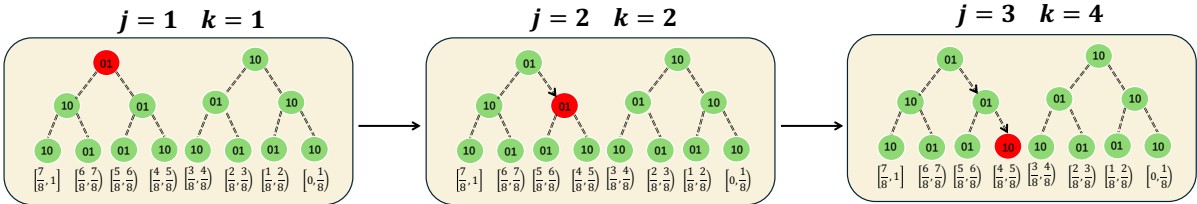

Figure 3: Illustration of Lines 7–9 in Algorithm 2 for the case $x^*_{\text{rank}} = 0.5$ and $m = 6$. The red circle highlights the $k$-th node at level $j$.

change the bin to which the row belongs. As a result, the recovered pseudorandom bits typically remain correct. Even when the statistic shifts noticeably, it usually moves only to an adjacent bin. Our node–bit mapping policy ensures that adjacent bins differ by only a single bit pair, which limits the effect of such shifts on the recovered bit sequence. The tree-based structure enables deterministic computation of these mappings without requiring explicit storage.

## 3    Analysis on Distortion and Robustness

This section provides a theoretical analysis of the distortion and robustness properties of our watermarking scheme. Throughout, we work in the transformed domain after YJT and standardization, where each entry is approximately normalized (zero mean and unit variance). This setting reduces feature heterogeneity and allows for a tractable analysis, which we formalize below.

**Assumption 1** (Centered and standardized transformed data)**.** *Let the tabular data* $\mathbf{X} = \{x_{i,j}\}_{i,j} \in \mathbb{R}^{N \times p}$ *be the output of the YJT and standardization pipeline. In particular, each column is centered and standardized:*

$$\sum_{i=1}^{N} x_{i,j} = 0 \text{ for all } j, \quad \text{and} \quad \mathbf{\Sigma} = \frac{1}{N}\mathbf{X}^\top \mathbf{X} \quad \text{with} \quad \text{diag}(\mathbf{\Sigma}) = \mathbb{I}_{p \times p}.$$

*Remark* 3 (Soundness of Assumption 1). Our analysis is conducted entirely in the transformed domain. Accordingly, we 1) omit the inverse YJT/standardization steps during watermark embedding and 2) do not refit transformation parameters (i.e., the YJT parameter $\lambda$ and the mean/variance used for standardization) during watermark detection. This idealization ignores the small distribution shifts in the watermarked frequency-domain representation induced by parameter refitting. This simplification is nevertheless sound in our setting because Tab-Drw preserves data fidelity well. Consequently, refitting the transformation parameters introduces only negligible distribution shift and has limited impact on the sign–bit alignment statistic used by our detector. Appendix D provides a detailed empirical justification (Tables 6 and 7) and additional discussion.

While the above assumption streamlines the theoretical development, all experiments in

Section 4 use the full watermark embedding and detection pipeline shown in Figure 1, consistent with realistic deployment. All technical proofs for this section are provided in Appendix E.

## 3.1 Watermark Distortion

Recall that each row vector $\boldsymbol{x}_i := (x_{i,0}, \ldots, x_{i,p-1}) \in \mathbb{R}^{1 \times p}$ is first mapped to the frequency domain via the DFT, then modified by adjusting a subset of imaginary components, and finally transformed back to the original domain. Under Assumption 1, Proposition 1 characterizes the resulting entry-wise distortion, i.e., the difference between the unwatermarked and watermarked table entries.

**Proposition 1** (Entry-wise differences). *Under Assumption 1, let $S \subseteq \{1, \ldots, m\}$ with $m = \lfloor \frac{p-1}{2} \rfloor$ denote the set of frequency coordinates whose imaginary signs are modified by our watermarking method. Let $\Delta x_{i,j} = x_{i,j}^{\mathrm{wm}} - x_{i,j}$ denote the entry-wise difference. Then*

$$\Delta x_{i,j} = -\alpha \, \boldsymbol{\beta}_j^\top \boldsymbol{x}_i, \quad \alpha = \frac{2(1 + \delta)}{p},$$

*where $\boldsymbol{\beta}_j = \left( \beta_S(0, j), \ldots, \beta_S(p-1, j) \right)^\top$, and $\beta_S(n, j) = \sum_{k \in S} \sin\left( \frac{2\pi k n}{p} \right) \sin\left( \frac{2\pi k j}{p} \right)$.*

Since TAB-DRW is primarily designed to watermark synthetic tabular data, we focus less on strict entry-wise similarity and more on preserving key statistical properties. In particular, we study how watermark embedding affects column-wise quantities, including the mean, inter-column correlations, and marginal distributions.

**Theorem 1** (Column-wise differences). *Under Assumption 1, our watermark affects column-wise quantities as follows:*

1. ***Mean.*** *For each column $j$, the column mean is preserved:*

$$\frac{1}{N} \sum_{i=1}^N \Delta x_{i,j} = 0.$$

2. ***Pearson correlation coefficients (PCC).*** *Let $r_{j\ell}$ and $r_{j\ell}^{\mathrm{wm}}$ be the PCCs between columns $j$ and $\ell$ before and after watermarking, respectively. Define $\Delta r_{j\ell} := r_{j\ell}^{\mathrm{wm}} - r_{j\ell}$. Then,*

$$\Delta r_{j\ell} = -\alpha \left( [\boldsymbol{\Sigma}\boldsymbol{\beta}_\ell]_j + [\boldsymbol{\Sigma}\boldsymbol{\beta}_j]_\ell \right) + \alpha^2 \boldsymbol{\beta}_j^\top \boldsymbol{\Sigma}\boldsymbol{\beta}_\ell.$$

3. ***Empirical distribution.*** *For each column $j$, let $\rho_j = \frac{1}{N} \sum_{i=1}^N \delta_{x_{i,j}}$ denote the empirical distribution of the unwatermarked entries, and let $\rho_j^{\mathrm{wm}} = \frac{1}{N} \sum_{i=1}^N \delta_{x_{i,j}^{\mathrm{wm}}}$ denote the corresponding distribution after watermarking. Let $\mathcal{W}_2(\cdot, \cdot)$ denote the Wasserstein-2 distance. Then,*

$$\mathcal{W}_2(\rho_j, \rho_j^{\mathrm{wm}}) \leq \alpha \sqrt{\boldsymbol{\beta}_j^\top \boldsymbol{\Sigma}\boldsymbol{\beta}_j}.$$

The above theorem characterizes column-wise differences and highlights how the parameters $(\gamma, \delta)$ influence them. When $\delta = -1$, no imaginary components are changed, so $\alpha = 0$ and all three column-wise quantities remain unchanged. Similarly, when $\gamma = 0$, the selected frequency set $S$ is empty, implying $\boldsymbol{\beta}_j = \mathbf{0}$ for all $j$, and again, no distortion occurs. In contrast, when $\gamma \neq 0$ and $\delta \neq -1$, the bounds quantify the extent of distortion, revealing how the watermark parameters affect the data.

## 3.2 Watermark Robustness

We now study the robustness of TAB-DRW under additive Gaussian noise. Our analysis is based on the Z-score in (3). Under the null hypothesis $H_0$ (unwatermarked data) and the standard independence assumption, the Z-score converges in distribution to $\mathcal{N}(0, 1)$ (see Lemma 3 in the appendix). In contrast, watermark embedding induces nontrivial sign–bit alignment in the frequency domain, leading to an elevated Z-score.

To make this intuition precise, we begin with a multivariate Gaussian model for the transformed data, formalized in Assumption 2. Theorem 2 then provides a closed-form lower bound on the expected Z-score computed from the noise-corrupted table, showing that the watermark signal remains detectable even under moderate Gaussian perturbations.

**Assumption 2** (Gaussian samples). *Let the unwatermarked tabular data $\mathbf{X} \in \mathbb{R}^{N \times p}$ have rows $\boldsymbol{x}_i \overset{\text{i.i.d.}}{\sim} \mathcal{N}(0, \boldsymbol{\Sigma})$, where $\boldsymbol{\Sigma} \in \mathbb{R}^{p \times p}$ is positive definite. Under an additive Gaussian-noise attack, the released table is $\mathbf{X}_{\text{wm}} + \boldsymbol{\varepsilon}$, where $\mathbf{X}_{\text{wm}}$ is watermarked with soft hyperparameters $(\gamma, \delta)$ and $\varepsilon_{i,j} \overset{\text{i.i.d.}}{\sim} \mathcal{N}(0, \sigma^2)$. Let the smallest and largest eigenvalues of $\boldsymbol{\Sigma}$ be $\lambda_{\min}$ and $\lambda_{\max}$, respectively, and denote $m = \lfloor \frac{p-1}{2} \rfloor$.*

**Theorem 2** (Robustness under Gaussian sample setting). *Under Assumption 2, define $Z(\gamma, \delta, \sigma)$ as the standard Z-score (as in (3)) computed on $\mathbf{X}_{\text{wm}} + \boldsymbol{\varepsilon}$, then for any $\gamma \in [0, 1]$ and $\delta, \sigma > 0$,*

$$\mathbb{E}\left[ Z(\gamma, \delta, \sigma) \right] \geq \sqrt{mN} \gamma \left[ 1 - \mathcal{I}(\sigma) - \mathcal{I}\left( \frac{\sigma}{\delta} \right) \right], \tag{4}$$

*where the function $\mathcal{I} : (0, \infty) \to \mathbb{R}$ is defined as*

$$\mathcal{I}(s) := \frac{s}{\sqrt{s^2 + \lambda_{\min}}} \left[ \Phi\left( \sqrt{1 + \frac{\lambda_{\min}}{s^2}} \right) - \frac{1}{2} \right] + \frac{s}{\sqrt{s^2 + \lambda_{\max}}} \left[ 1 - \Phi\left( \sqrt{1 + \frac{\lambda_{\max}}{s^2}} \right) \right] + \frac{1}{\sqrt{8\pi e}} \left[ E_1\left( \frac{\lambda_{\min}}{2s^2} \right) - E_1\left( \frac{\lambda_{\max}}{2s^2} \right) \right].$$

*with $\Phi(\cdot)$ denoting the standard normal CDF and $E_1(u) = \int_u^\infty \frac{e^{-t}}{t} dt$ the exponential integral.*

Based on the Assumption 2, we also derived a closed-form lower bound on the number of rows $N$ required to achieve a one-sided statistical test with power $1 - \beta$ at significance level $\alpha$. Corollary 1 below gives a formal description.

**Corollary 1** (Sample-size lower bound). *Under Assumption 2, define $N_{\alpha, \beta}(\gamma, \delta, \sigma)$ as the number of rows required for the Gaussian noise-corrupted table $\mathbf{X}_{\text{wm}} + \boldsymbol{\varepsilon}$ to achieve a one-sided test with*

*power $1 - \beta$ at significance level $\alpha$. Then, for any $\gamma, \alpha, \beta \in [0,1]$ and $\delta, \sigma > 0$,*

$$N_{\alpha,\beta}(\gamma, \delta, \sigma) \geq \frac{\left[ q_\alpha + \sqrt{2m \ln \left( \frac{1}{\beta} \right)} \right]^2}{m\gamma^2 \left[ 1 - \mathcal{I}(\sigma) - \mathcal{I}(\frac{\sigma}{\delta}) \right]^2}, \tag{5}$$

*where $q_\alpha$ is the critical value for a one-sided test at level $\alpha$.*

*Remark* 4 (Numerical illustration). To illustrate the lower bound in Theorem 2, we present a simple numerical example. Consider $(N, p) = (1000, 11)$ with $\mathbf{\Sigma} = \mathbb{I}_{p \times p}$. Then the right-hand side of (4) yields

$$\mathbb{E}\left[ Z(0.5, 0.5, \sigma) \right] \geq \begin{cases} 30.13, & \text{if } \sigma = 0.1, \\ 14.95, & \text{if } \sigma = 0.5, \\ 7.04, & \text{if } \sigma = 1.0. \end{cases}$$

Since the 0.99 quantile of a standard normal is 2.32, these values indicate that the watermark signal remains significant even under moderate noise corruption.

Moreover, Corollary 1 provides an explicit sample-size requirement. In the same setting with $p = 11$ (so $m = 5$), achieving a one-sided test with power $1 - \beta = 0.99$ at level $\alpha = 0.001$ (i.e., 0.99 TPR@0.1%FPR) requires

$$N_{0.001, 0.01}(0.5, 0.5, \sigma) \geq \begin{cases} 108, & \text{if } \sigma = 0.1, \\ 153, & \text{if } \sigma = 0.2, \\ 437, & \text{if } \sigma = 0.5. \end{cases}$$

Since real-world tabular data are highly heterogeneous and rarely follow a strict multivariate Gaussian distribution, even after YJT and standardization, we further extend the robustness analysis beyond the Gaussian sample setting to a sub-Gaussian setting in Theorem 3. This setting accommodates light-tailed, non-Gaussian features, including bounded or discrete categorical variables.

**Assumption 3** (Sub-Gaussian samples). *Let the unwatermarked tabular data $\mathbf{X} \in \mathbb{R}^{N \times p}$ have i.i.d. rows $\boldsymbol{x_i} \in \mathbb{R}^p$ with zero mean and covariance $\mathbf{\Sigma} \in \mathbb{R}^{p \times p}$. Moreover, $\boldsymbol{x_i}$ is $\mathbf{\Sigma}$–sub-Gaussian: there exists $\kappa \geq 1$ such that for every $u \in \mathbb{R}^p$, $\| \langle u, \boldsymbol{x_i} \rangle \|_{\psi_2} \leq \kappa \sqrt{u^\top \mathbf{\Sigma} u}$ (Vershynin, 2018). Under an additive Gaussian-noise attack, the released table is $\mathbf{X}_{\mathrm{wm}} + \boldsymbol{\varepsilon}$, where $\mathbf{X}_{\mathrm{wm}}$ is watermarked with soft hyperparameters $(\gamma, \delta)$ and $\varepsilon_{i,j} \overset{\mathrm{i.i.d.}}{\sim} \mathcal{N}(0, \sigma^2)$. Denote the smallest eigenvalue of $\mathbf{\Sigma}$ by $\lambda_{\min}$, and let $m = \lfloor \frac{p-1}{2} \rfloor$.*

**Theorem 3** (Robustness under sub-Gaussian sample setting). *Under Assumption 3, define $Z(\gamma, \delta, \sigma)$ as the standard Z-score (as in (3)) computed on $\mathbf{X}_{\mathrm{wm}} + \boldsymbol{\varepsilon}$. Fix any $\theta \in (0, 1)$ and*

*let* $C_4 > 0$ *denote a constant such that for every real sub-Gaussian $U$ with variance $v$ we have* $\mathbb{E}U^4 \leq C_4\,\kappa^4\,v^2$. *Define* $\rho(\kappa, \theta) := \frac{(1-\theta)^2}{2\,C_4\,\kappa^4}$, *then for any $\gamma \in [0,1]$ and $\delta, \sigma > 0$,*

$$\mathbb{E}\big[Z(\gamma, \delta, \sigma)\big] \;\geq\; \sqrt{mN}\,\gamma \sup_{\theta \in (0,1)} \left\{ \rho(\kappa, \theta) \left[ 2 - \exp\Big( -\frac{\theta\,\lambda_{\min}}{2\,\sigma^2} \Big) - \exp\Big( -\frac{\theta\,\lambda_{\min}\,\delta^2}{2\,\sigma^2} \Big) \right] \right\}.$$

*Remark* 5 (What the sub-Gaussian setting covers). The $\mathbf{\Sigma}$-sub-Gaussian setting strictly generalizes the Gaussian setting used in Theorem 2 and many non-Gaussian settings that are common in tabular data: 1) bounded/quantized/discrete features (e.g., `gender` or `education` features), and 2) finite mixtures of light-tailed distributions with uniformly bounded covariances (a finite mixture of sub-Gaussians is sub-Gaussian with the worst-component parameter). In conclusion, non-Gaussian distribution with light tails are covered.

## 4 Experiments

In this section, we empirically evaluate TAB-DRW on five benchmark tabular datasets. We begin with the experimental setup in Section 4.1, then study 1) data fidelity and watermark detectability, including the trade-off induced by the choice of soft hyperparameters, in Section 4.2, 2) robustness against post-processing attacks in Section 4.3, 3) robustness against adaptive attacks in Section 4.4, and 4) watermark embedding and detection runtime in Section 4.5. Finally, We present a case study examining the impact of watermark embedding on low-entropy categorical variables in Section 4.6. Code is available at https://github.com/zhyzmath/TAB-DRW-Tabular-Data-Watermarking.

### 4.1 Experimental Setup

**Datasets.** Experiments are conducted on five real-world tabular datasets with both continuous and discrete feature types: Adult (Becker and Kohavi, 1996), Magic (Bock, 2004), Shoppers (Sakar and Kastro, 2018), Default (Yeh, 2009), and Drybean (Koklu and Özkan, 2020). See details in Appendix F.1.

**Baselines.** We consider four baselines: two post-editing watermarking methods, GLW (He et al., 2024) and TabularMark (Zheng et al., 2024), and two sampling-phase methods, TabWak* with valid bit mechanism (Zhu et al., 2025) and MUSE (Fang et al., 2025). We reproduce TabWak* using the official open-source code and implement other baselines according to the authors' specifications. To ensure fair comparison, we follow Zhu et al. (2025) to synthesize tabular data using TabSyn (Zhang et al., 2024a) with DDIM sampling. Further implementation details are provided in Appendix F.3 & F.4.

**Metrics.** We evaluate data fidelity using four metrics introduced in Zhang et al. (2024a): 1) **Density**, which quantifies column-wise distributional similarity between synthetic and real data, 2) **Corr**, which measures how well inter-column correlations are preserved, 3) **C2ST**, the classifier

Table 2: Data fidelity and watermark detectability. No watermarking is denoted as "W/O". Our proposed Tab-Drw is evaluated with the hyperparameter $(\gamma, \delta) = (0.5, 0.5)$. Best performances are shown in **bold**, and second-best are underlined.

| Datasets | Method | Fidelity Metric | | | | Z-score | | FPR / TPR | |
|---|---|---|---|---|---|---|---|---|---|
| | | Density ↑ | Corr ↑ | C2ST ↑ | MLE ↑ | 1K rows ↑ | 5K rows ↑ | 1K rows ↑ | 5K rows ↑ |
| Adult | W/O | 0.922±0.001 | 0.872±0.001 | 0.611±0.004 | 0.824±0.005 | – | – | – | – |
| | GLW | 0.912±0.002 | 0.871±0.002 | 0.604±0.015 | 0.821±0.017 | 7.293±0.96 | 16.54±1.05 | 0.00/0.91 | 0.00/1.00 |
| | MUSE | 0.921±0.001 | **0.877±0.003** | 0.599±0.007 | 0.823±0.005 | 6.712±0.89 | 14.81±1.23 | 0.00/0.78 | 0.00/1.00 |
| | TabWak* | 0.912±0.002 | 0.863±0.004 | 0.604±0.009 | 0.793±0.009 | 6.796±1.03 | 15.67±0.97 | 0.00/0.78 | 0.00/1.00 |
| | TabularMark | **0.922±0.001** | 0.872±0.001 | 0.598±0.006 | **0.823±0.003** | 9.674±3.00 | 22.53±3.05 | 0.00/0.89 | 0.00/1.00 |
| | **TAB-DRW** | 0.915±0.005 | 0.864±0.004 | **0.604±0.008** | 0.816±0.009 | **12.81±1.17** | **29.55±1.12** | 0.00/1.00 | 0.00/1.00 |
| Magic | W/O | 0.917±0.001 | 0.945±0.003 | 0.672±0.004 | 0.823±0.007 | – | – | – | – |
| | GLW | 0.915±0.001 | **0.944±0.002** | 0.669±0.013 | 0.816±0.006 | **77.05±0.55** | **172.2±0.51** | 0.00/1.00 | 0.00/1.00 |
| | MUSE | 0.912±0.002 | 0.943±0.006 | 0.672±0.008 | 0.824±0.017 | 15.84±0.86 | 35.31±0.70 | 0.00/1.00 | 0.00/1.00 |
| | TabWak* | 0.912±0.007 | 0.921±0.003 | 0.671±0.006 | **0.827±0.029** | 8.608±0.98 | 19.83±1.01 | 0.00/1.00 | 0.00/1.00 |
| | TabularMark | **0.917±0.001** | 0.943±0.003 | 0.674±0.005 | 0.822±0.009 | 9.666±2.82 | 22.02±3.62 | 0.00/0.90 | 0.00/1.00 |
| | **TAB-DRW** | 0.917±0.005 | 0.937±0.003 | **0.676±0.009** | 0.818±0.014 | 27.34±0.93 | 61.42±1.02 | 0.00/1.00 | 0.00/1.00 |
| Shoppers | W/O | 0.919±0.002 | 0.910±0.001 | 0.704±0.005 | 0.902±0.012 | – | – | – | – |
| | GLW | 0.903±0.001 | **0.908±0.001** | 0.706±0.018 | 0.893±0.009 | 17.84±1.12 | 39.08±1.05 | 0.00/1.00 | 0.00/1.00 |
| | MUSE | 0.911±0.001 | 0.908±0.002 | 0.710±0.009 | 0.895±0.012 | 12.80±0.88 | 28.83±0.87 | 0.00/1.00 | 0.00/1.00 |
| | TabWak* | **0.916±0.009** | 0.906±0.001 | 0.674±0.021 | **0.905±0.047** | 4.071±1.06 | 10.38±1.02 | 0.00/0.04 | 0.00/1.00 |
| | TabularMark | 0.914±0.003 | 0.908±0.001 | 0.704±0.005 | 0.897±0.018 | 10.28±3.24 | 22.94±3.36 | 0.00/0.91 | 0.00/1.00 |
| | **TAB-DRW** | 0.909±0.006 | 0.902±0.003 | **0.712±0.013** | 0.891±0.014 | **18.18±1.28** | **40.74±1.26** | 0.00/1.00 | 0.00/1.00 |
| Default | W/O | 0.930±0.001 | 0.907±0.001 | 0.717±0.003 | 0.797±0.009 | – | – | – | – |
| | GLW | 0.926±0.002 | 0.906±0.003 | 0.710±0.027 | 0.787±0.011 | 12.10±1.09 | 27.08±0.99 | 0.00/1.00 | 0.00/1.00 |
| | MUSE | 0.928±0.001 | 0.907±0.002 | 0.714±0.008 | 0.790±0.007 | 15.50±0.97 | 34.42±0.95 | 0.00/1.00 | 0.00/1.00 |
| | TabWak* | **0.934±0.011** | **0.912±0.014** | **0.723±0.048** | 0.775±0.009 | 10.49±1.03 | 23.60±1.00 | 0.00/1.00 | 0.00/1.00 |
| | TabularMark | 0.927±0.005 | 0.902±0.006 | 0.718±0.007 | **0.796±0.009** | 9.526±2.91 | 23.94±3.18 | 0.00/0.89 | 0.00/1.00 |
| | **TAB-DRW** | 0.929±0.010 | 0.907±0.011 | 0.717±0.018 | 0.791±0.013 | **15.98±0.92** | **35.84±0.91** | 0.00/1.00 | 0.00/1.00 |
| Drybean | W/O | 0.932±0.001 | 0.935±0.001 | 0.640±0.003 | 0.878±0.009 | – | – | – | – |
| | GLW | 0.929±0.002 | 0.933±0.004 | 0.637±0.017 | 0.872±0.013 | **55.14±0.66** | **123.3±0.68** | 0.00/1.00 | 0.00/1.00 |
| | MUSE | 0.930±0.003 | 0.934±0.005 | 0.649±0.011 | 0.878±0.014 | 14.14±1.03 | 31.43±0.91 | 0.00/1.00 | 0.00/1.00 |
| | TabWak* | 0.924±0.014 | 0.925±0.018 | **0.659±0.032** | 0.875±0.015 | 7.999±0.92 | 17.80±0.97 | 0.00/0.99 | 0.00/1.00 |
| | TabularMark | **0.932±0.002** | **0.935±0.001** | 0.641±0.005 | 0.878±0.011 | 7.760±3.14 | 17.28±2.73 | 0.00/0.71 | 0.00/1.00 |
| | **TAB-DRW** | 0.931±0.013 | 0.928±0.007 | 0.655±0.029 | **0.880±0.019** | 38.03±1.03 | 85.05±0.67 | 0.00/1.00 | 0.00/1.00 |

two-sample test score measuring distinguishability between synthetic and real data, 4) **MLE**, the performance of downstream models trained on synthetic data. For watermark detection, we report two statistical metrics: the one-sided **Z-score** defined in (3), which quantifies the distributional shift of a pivotal statistic induced by watermarking, and **FPR / TPR**, the false and true positive rates under the critical value $q_\alpha = 6$, chosen to better differentiate detection performance. For fair comparison across baselines, we convert each method's detector into the unified one-sided Z-score based on its row-level statistic. The only exception is TabularMark, for which we retain its original cell-based detector. See Appendix F.2 for further details.

## 4.2 Data Fidelity vs. Watermark Detectability

To evaluate data fidelity, we generate synthetic tabular datasets with the same number of rows as the original for each of the five datasets. For watermark detectability, we compute both Z-scores and FPR/TPR using two batch sizes: 1K and 5K rows. Table 2 reports the mean and standard deviation of each fidelity metric over 10 independent trials, and each detectability metric over 100 trials.

On one hand, our watermarking scheme introduces **minimal distortion**. Across all datasets, its fidelity scores closely match those of the unwatermarked data (degrading by no more than 0.01) and are comparable to existing baselines. While TabWak* and TabularMark often achieve the highest fidelity, our method with $(\gamma, \delta) = (0.5, 0.5)$ performs similarly well. Overall, all baseline methods yield comparable fidelity scores, indicating that our approach preserves data

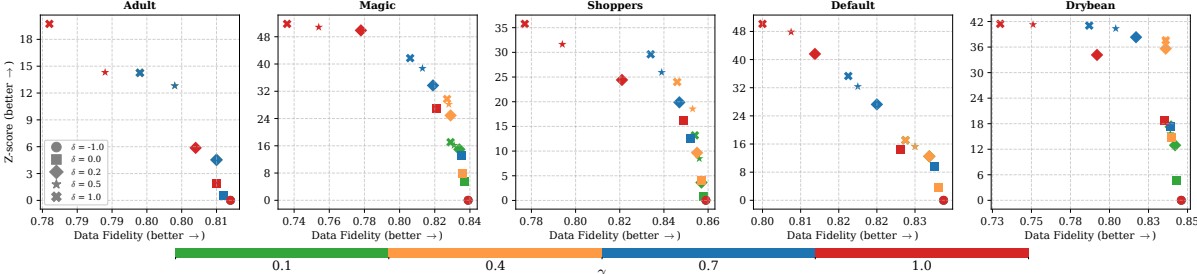

Figure 4: Trade-off between average Z-score on 1K-rows tables and data fidelity under varying $(\gamma, \delta)$.

quality on par with prior work. On the other hand, in terms of watermark detectability, our method achieves the **highest Z-scores** on the Adult, Shoppers, and Default datasets. GLW performs best on Magic and Drybean, likely due to the predominance of continuous attributes. Most methods—including ours—successfully control false positive rates and show nontrivial true positive rates under the critical threshold $q_\alpha = 6$, demonstrating reliable detectability.

The results in Figure 4 also reveal a trade-off between data fidelity and watermark strength for our method. Increasing both $\gamma$ and $\delta$ enhances the Z-score, but also increases distortion. This trade-off is inherent to post-editing watermarking: stronger signals inevitably introduce more distortion. Similar additional results on TabSyn with score-based diffusion process and two other tabular generators, together with broader ablation study, are presented in Appendix G.1.

## 4.3   Robustness against Common Post-processing Attacks

We next evaluate the robustness of watermarking methods against ten post-processing attacks, which can be grouped into four categories: 1) deletion attacks, which remove or replace information at different granularities (**Row Del.**, **Col Del.**, **Cell Del.**), 2) noise attacks, which perturb numerical or categorical values with Gaussian, categorical, or adaptive noise (**G-Noise**, **C-Noise**, **A-Noise**), 3) discretization attacks, which reduce numeric precision through truncation or quantization (**Truncation**, **Quantization**), and 4) structural attacks, which alter table structure by resampling label distributions or randomly shuffling rows (**Resample**, **Shuffle**). We do not include column-level shuffling in our attack suite because the original column order can typically be accurately recovered by using headers, basic statistical properties (e.g., mean and standard deviation), or semantic features of each column. These characteristics are usually distinctive enough to make column permutation easily reversible. This setting is also standard in prior works (Zhu et al., 2025; Fang et al., 2025). Detailed definition and implementations of the attacks are provided in Appendix F.5, and additional results of stronger attack intensities are reported in Appendix G.2.

Table 3 reports the average one-sided *Z*-score over 5K rows. Our watermarking method **consistently demonstrates superior robustness across all attack types and datasets**, ranking either first or second. In contrast, GLW and TabularMark remain resilient to deletion and structural attacks but often fail under noise and discretization. TabWak* and MUSE exhibit some robustness to noise and discretization on certain datasets, but they are vulnerable to deletion attacks due to their reliance on complete column information.

Table 3: Watermark robustness against common attacks. Average Z-score on 5K rows under ten post-processing attacks. Each value is obtained by repeating the attacks 100 times (10 times for "TabWak*") and averaging the results. Our proposed TAB-DRW is evaluated with the hyperparameter $(\gamma, \delta) = (0.5, 0.5)$. Best performances are shown in **bold**, and second-best are underlined.

| Datasets | Methods | Attacks | | | | | | | | | |
|---|---|---|---|---|---|---|---|---|---|---|---|
| | | Row Del. | Col Del. | Cell Del. | G-Noise | C-Noise | A-Noise | Truncation | Quantization | Resample | Shuffle |
| Adult | GLW | 15.69 | 14.55 | 14.88 | 0.00 | 16.54 | 7.26 | 16.54 | 8.63 | 16.89 | 16.54 |
| | MUSE | 14.00 | 6.26 | 9.16 | 12.83 | 10.91 | 4.53 | 14.81 | 10.96 | 20.15 | 14.81 |
| | TabWak* | 14.98 | 11.32 | 11.08 | 0.91 | 15.67 | 14.50 | 5.09 | 11.27 | 16.37 | 15.67 |
| | TabularMark | 21.65 | 15.56 | 16.42 | 2.83 | 6.90 | 1.29 | 2.72 | 0.00 | 4.62 | 17.44 |
| | **TAB-DRW** | **27.98** | **17.78** | **20.46** | **20.36** | **24.59** | **23.72** | **29.55** | **20.95** | **28.15** | **29.55** |
| Magic | GLW | **163.33** | **140.15** | **154.89** | 0.03 | **172.20** | 1.09 | 14.71 | **47.28** | **170.81** | **172.20** |
| | MUSE | 33.55 | 7.16 | 17.91 | 15.99 | 34.33 | 9.09 | 20.06 | 14.54 | 33.59 | 35.31 |
| | TabWak* | 18.92 | 13.18 | 16.33 | 17.99 | 19.76 | 13.86 | 16.87 | 16.50 | 17.62 | 19.76 |
| | TabularMark | 20.65 | 12.05 | 13.80 | 0.00 | 20.05 | 0.35 | 13.66 | 0.65 | 21.04 | 15.26 |
| | **TAB-DRW** | 58.28 | 17.45 | 35.78 | **46.18** | 54.48 | **40.72** | **52.62** | 45.14 | 37.61 | 61.42 |
| Shoppers | GLW | 36.82 | **36.02** | **36.15** | 0.00 | **39.08** | 1.11 | 24.61 | 7.20 | **31.92** | 39.08 |
| | MUSE | 27.34 | 11.37 | 15.34 | 23.56 | 21.58 | 16.74 | 23.14 | 19.84 | 28.63 | 28.83 |
| | TabWak* | 9.55 | 3.33 | 4.68 | 0.02 | 10.47 | 5.08 | 9.05 | 7.82 | 10.71 | 10.47 |
| | TabularMark | 15.48 | 11.17 | 14.66 | 1.46 | 18.43 | 0.13 | 11.40 | 3.02 | 18.23 | 18.62 |
| | **TAB-DRW** | **38.43** | 19.35 | 22.99 | **39.66** | 36.26 | **20.66** | **30.28** | **32.46** | 29.28 | **40.74** |
| Default | GLW | 25.67 | 24.59 | **23.88** | 0.00 | 27.08 | 9.08 | 27.08 | 11.94 | 15.82 | 27.08 |
| | MUSE | 32.75 | 11.38 | 13.11 | 19.81 | 24.25 | 4.98 | 34.42 | 11.17 | **36.60** | 34.42 |
| | TabWak* | 22.91 | 18.36 | 18.96 | 22.95 | 23.70 | **21.79** | 22.80 | 19.83 | 31.49 | 23.70 |
| | TabularMark | 21.66 | 15.79 | 12.46 | 0.00 | 21.53 | 0.23 | 12.36 | 0.86 | 20.94 | 23.33 |
| | **TAB-DRW** | **33.92** | **25.03** | 22.56 | **30.03** | **32.22** | 21.55 | **35.84** | **21.93** | 32.36 | **35.84** |
| Drybean | GLW | **116.96** | **112.05** | **110.89** | 0.00 | **123.28** | 13.29 | 28.02 | 29.37 | **123.68** | **123.28** |
| | MUSE | 29.78 | 7.76 | 12.32 | 6.22 | 29.56 | 6.28 | 10.43 | 1.89 | 31.19 | 31.43 |
| | TabWak* | 17.16 | 0.00 | 2.86 | 14.11 | 17.53 | 13.59 | 16.80 | 6.56 | 15.38 | 17.53 |
| | TabularMark | 13.79 | 7.18 | 9.55 | 0.00 | 13.56 | 0.00 | 7.57 | 0.66 | 12.88 | 10.46 |
| | **TAB-DRW** | 80.62 | 42.82 | 50.99 | **31.12** | 80.43 | **58.50** | **42.14** | **61.23** | 68.69 | 85.05 |

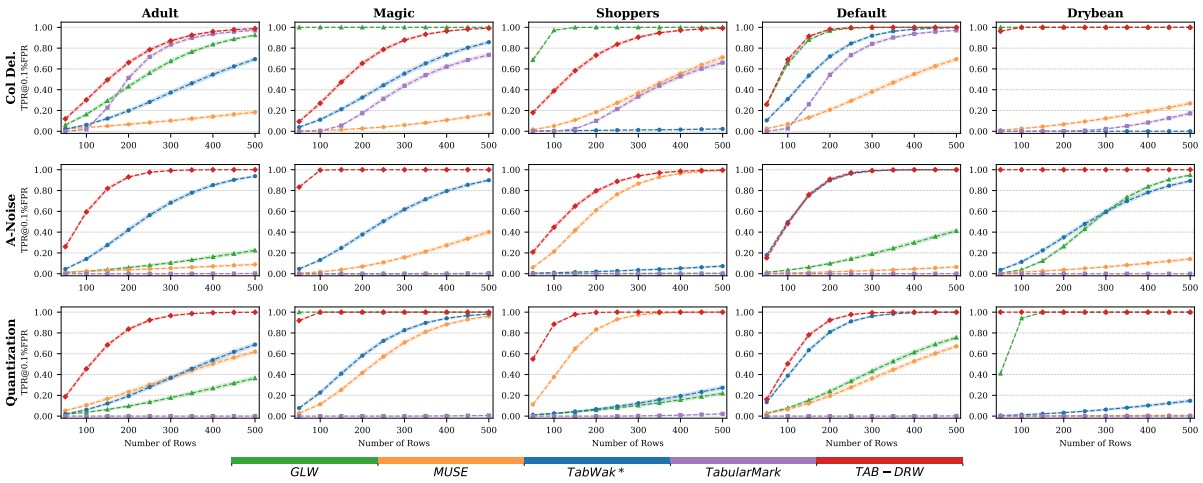

Figure 5: TPR@0.1%FPR versus row count under three representative attacks. Dashed lines show the bootstrap mean estimate (500 resamples), and shaded regions indicate the 90% confidence interval.

Figure 5 further highlights the sample efficiency of our method by plotting TPR@0.1%FPR against the number of rows under three representative and strong attacks. In ten out of fifteen cases, our method achieves perfect detection (TPR = 1.0) using only 300 rows, while the remaining five cases require fewer than 500 rows. In contrast, baseline methods often experience substantial drops in true positive rate or lose detectability entirely under the same conditions.

Table 4: Robustness of TAB-DRW against adaptive row deletion and rewatermarking attacks of varying strength. Z-scores are computed on tables with 5K rows and averaged over 100 independent trials. "Rewatermarking@$n$" denotes rewatermarking the table using $n$ randomly sampled keys.

| Datasets | No-attack | Adv. Row Del. | | | Rewatermarking | | |
|---|---|---|---|---|---|---|---|
| | | @0.1 | @0.2 | @0.5 | @1 | @3 | @10 |
| Adult | $29.12 \pm 1.12$ | $28.55 \pm 1.44$ | $26.35 \pm 2.61$ | $18.79 \pm 4.47$ | $23.66 \pm 1.17$ | $16.26 \pm 1.09$ | $17.26 \pm 1.34$ |
| Magic | $61.42 \pm 1.02$ | $56.71 \pm 2.98$ | $49.36 \pm 5.77$ | $28.41 \pm 7.28$ | $53.23 \pm 0.91$ | $34.32 \pm 0.93$ | $29.17 \pm 1.00$ |
| Shoppers | $40.74 \pm 1.26$ | $36.47 \pm 2.62$ | $30.40 \pm 3.71$ | $17.12 \pm 4.18$ | $31.97 \pm 1.15$ | $20.14 \pm 1.09$ | $16.67 \pm 1.09$ |
| Default | $35.84 \pm 0.91$ | $31.91 \pm 1.90$ | $27.13 \pm 3.23$ | $15.36 \pm 4.54$ | $32.85 \pm 1.00$ | $19.40 \pm 1.07$ | $26.28 \pm 1.18$ |
| Drybean | $85.05 \pm 0.67$ | $79.27 \pm 2.67$ | $71.67 \pm 5.04$ | $52.39 \pm 9.99$ | $44.79 \pm 0.81$ | $29.47 \pm 0.83$ | $33.77 \pm 0.95$ |

## 4.4 Robustness against Adaptive Attacks

In this section, we implement two adaptive attacks specifically targeting TAB-DRW to evaluate robustness under stronger adversarial settings. In both cases, we assume adversaries have full knowledge of the watermarking pipeline, including the privacy-enhanced variant used for real-world deployment (see Appendix B), but do not have access to the secret key.

The first attack, **Adaptive Row Deletion**, corrupts the row ranking to impair rank-based bit retrieval. The attacker samples a random key, computes normalized row ranks (following lines 3–6 of Algorithm 2), and deletes a contiguous block of rows in rank space. For example, with strength 0.1, the adversary removes rows whose normalized ranks lie in a random interval of length 0.1 in $[0, 1]$, which disrupts ranking far more than random deletion. The second attack, **Rewatermarking**, aims to erase sign-bit alignment in the frequency domain. It leverages two properties of TAB-DRW: strong fidelity preservation and the fact that a watermark embedded with one key is not detectable by another (See Table 21 in Appendix G.3 for empirical justification). An informed attacker can repeatedly rewatermark the table with different keys to perturb the original alignment and make detection under the original key fail.

The results in Table 4 show that TAB-DRW remains **highly detectable even under substantial adaptive row-deletion attacks**. Although detectability decreases slightly relative to random row deletion, the use of a secret key and stable tree-based bit storage makes TAB-DRW resilient to attacks specifically designed to disrupt the row-ranking process. We also observe that TAB-DRW remains highly detectable even after ten rounds of rewatermarking, by which point tabular fidelity has already degraded noticeably (cf. Table 18). These findings demonstrate that, without knowledge of the key used in Algorithm 2 and 3, an attacker—despite understanding the TAB-DRW pipeline—cannot substantially disrupt sign-bit alignment while preserving data fidelity.

## 4.5 Runtime Evaluation

Table 5 reports the average runtimes for watermark embedding and detection on five benchmark datasets. Each result is averaged over 100 independent trials on synthetic tabular data with 1K rows. All experiments are conducted on an M1 Pro CPU and a 40GB NVIDIA A100 GPU.

From a deployment perspective, two complementary facts are worth emphasizing. First, watermark embedding is a one-time post-processing step associated with releasing a watermarked table. At the 1K-row scale of Table 5, producing the underlying synthetic table in our pipeline

Table 5: Runtime comparison across watermarking methods on 1K-row tables (in seconds). Each cell reports **Embedding / Detection** time. Values for TabWak* denote GPU runtime, whereas all other methods are measured in CPU time.

| Dataset | GLW | | MUSE | | TabWak* | | TabularMark | | TAB-DRW | |
|---|---|---|---|---|---|---|---|---|---|---|
| | **Embed** | **Detect** | **Embed** | **Detect** | **Embed** | **Detect** | **Embed** | **Detect** | **Embed** | **Detect** |
| Adult | 0.031 | 0.003 | 0.593 | 0.177 | 0.008 | 27.96 | 0.008 | 0.717 | 0.112 | 0.100 |
| Magic | 0.026 | 0.001 | 0.555 | 0.161 | 0.004 | 21.78 | 0.007 | 0.729 | 0.106 | 0.076 |
| Shoppers | 0.026 | 0.005 | 0.629 | 0.188 | 0.020 | 30.27 | 0.008 | 0.682 | 0.142 | 0.106 |
| Default | 0.052 | 0.004 | 0.688 | 0.234 | 0.020 | 35.49 | 0.009 | 0.713 | 0.205 | 0.152 |
| Drybean | 0.025 | 0.003 | 0.601 | 0.193 | 0.016 | 26.03 | 0.009 | 0.582 | 0.152 | 0.120 |

takes roughly 2 seconds GPU time, so the additional embedding overhead of GLW, TabularMark, TabWak*, and TAB-DRW is negligible relative to generation. The only notable exception is MUSE, which is more expensive because it must generate multiple candidate rows and then select among them. Second, detection is often performed repeatedly in practice. For example, auditors or downstream users may examine multiple suspect tables, and this becomes even more frequent in multi-key deployment scenarios where the same table is checked against several candidate keys. In such settings, detection efficiency becomes the more critical factor. This further highlights the practical advantage of TAB-DRW, particularly compared to TabWak*, whose detector relies on computationally expensive DDIM inversion, and TabularMark, which consistently incurs higher detection costs due to tuple matching.

Finally, we note that once unwatermarked samples are generated, the grid-search tuning procedure described in Section 2 can be applied to the synthetic data with negligible overhead—on the order of seconds for datasets of comparable scale to our benchmarks—and can also be carried out entirely on CPU.

## 4.6 A Case Study on Low-Entropy Categorical Variable

TAB-DRW handles low-entropy categorical variables (e.g., `gender`) in a conservative and adaptive manner. Because watermark embedding is performed in the frequency domain on a row-wise DFT representation, each standardized sample is mapped to a joint representation whose components are linear combinations of all features in the row. Consequently, watermark embedding via imaginary sign-bit alignment operates on this joint representation rather than on any individual attribute. Whether a sample's `gender` value remains unchanged or flips therefore depends on its overall position relative to the distribution of other features, rather than on the `gender` attribute alone.

To illustrate this behavior empirically, we conduct a case study on the Adult dataset, focusing on potential flips of the `gender` variable. We randomly select three pairs of synthetic 5K-row tables (unwatermarked & watermarked). After applying the YJT to standardize feature scales on the unwatermarked data, we perform principal component analysis (PCA) on all non-`gender` variables and retain the first two principal components. Based on these components, we apply $k$-means clustering with $k = 2$. We label the cluster containing a higher proportion of samples originally labeled as `Male` as the `Male` cluster, and assign the remaining cluster the `Female` label.

We then examine samples whose `gender` value flips after watermarking and partition them

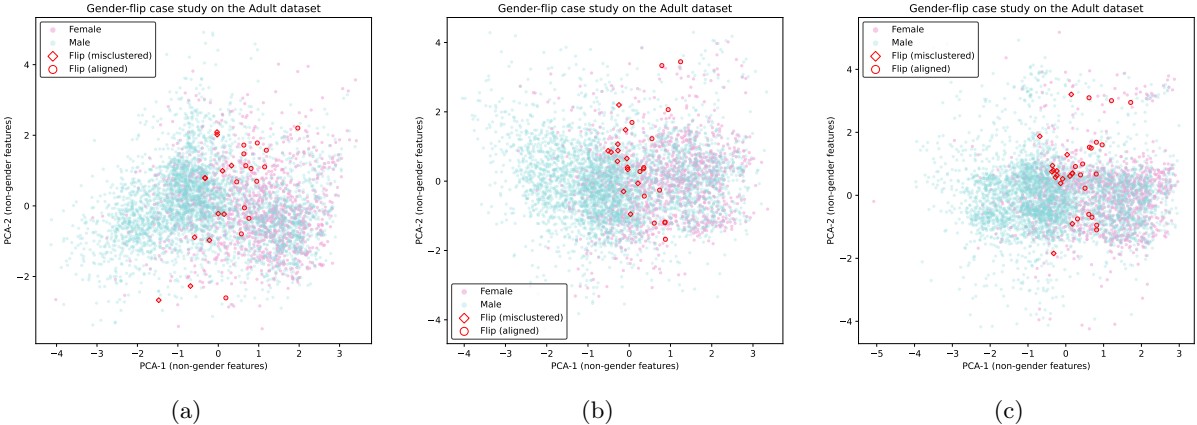

Figure 6: Visualization of the `gender`-flipping case study on the Adult dataset. Each subfigure corresponds to a synthetic 5K-row table pair (unwatermarked & watermarked). Samples whose `gender` value flips after watermarking are highlighted. "Flip (misclustered)" denotes samples whose original `gender` label conflicts with their cluster label prior to watermarking, while "Flip (aligned)" denotes samples whose original label agrees with the cluster label but still flips. In subfigure (a), 26 out of 5K samples exhibit a `gender` flip (46.2% misclustered), with an average distance to the cluster boundary of 0.31, compared with 0.78 for the remaining samples. In subfigure (b), 27 out of 5K samples flip (48.1% misclustered), with an average boundary distance of 0.26 versus 0.75 for the remaining samples. In subfigure (c), 33 out of 5K samples flip (48.5% misclustered), with an average boundary distance of 0.29 versus 0.77 for the remaining samples.

into two groups: *aligned* samples, whose original `gender` label agrees with the assigned cluster label, and *misclustered* samples, whose original label conflicts with the cluster assignment. To further evaluate whether flips concentrate on samples that are intrinsically ambiguous with respect to the two clusters, and thus avoid semantic distortion, we quantify each sample's proximity to the separation between clusters. Specifically, we define the *cluster boundary* as the set of points in the PCA space that are equidistant (in Euclidean distance) from the two cluster centroids, and define the *boundary distance* of a sample as its shortest Euclidean distance to this boundary. We use this metric to compare flipped and non-flipped samples and to localize where flips occur in the PCA space. Combined with the visualization in Figure 6, it reveals several consistent patterns:

1. Flips in low-entropy categorical variables are rare under TAB-DRW. In a 5K-row table, we typically observe only around 30 such cases.

2. Nearly half of the flipped samples are already misclustered before watermarking, indicating that the embedding process does not introduce semantic inconsistency and may even improve alignment with other features.

3. For the remaining flipped samples whose original labels align with their assigned cluster labels, the samples tend to lie close to the cluster boundary. Across the three table pairs, their average boundary distance is 0.29, compared with 0.77 for the remaining samples, suggesting that the resulting flips are confined to semantically ambiguous regions.

Overall, this case study demonstrates that TAB-DRW **adapts its effective embedding behavior to feature entropy, producing conservative and semantics-preserving modifications for low-entropy categorical variables**.

# 5   Discussion

In this paper, we present TAB-DRW, a lightweight and robust post-editing watermarking scheme for tabular data. TAB-DRW normalizes heterogeneous features via the Yeo–Johnson transformation and standardization, and embeds watermarks by adjusting the imaginary parts of adaptively selected frequency-domain entries. It achieves strong detectability and high fidelity across mixed-type datasets—without relying on large diffusion models or explicitly storing unwatermarked data. Our proposed rank-based pseudorandom bit generation method enables efficient row-wise retrieval via robust rank statistics, further enhancing resilience to post-processing attacks. We provide theoretical analysis of watermark distortion and robustness against noise perturbations, and validate our approach on five benchmark datasets, demonstrating broad applicability, high fidelity, strong detectability, and great robustness against common post-processing attacks and stronger adaptive attacks. We believe that TAB-DRW offers a solid foundation for advancing secure data sharing and the development of privacy-preserving generative AI.

Several directions remain open for future work. First, it is worth exploring whether there exists a provably optimal strategy for modifying the frequency-domain representation to maximize detectability while minimizing distortion—and if so, in what sense. Second, integrating TAB-DRW with differential privacy or membership inference protections could provide unified mechanisms for data traceability and privacy preservation. Third, adapting the watermark strength based on feature importance or downstream task performance could further improve the fidelity-detectability trade-off. We hope these directions inspire further research toward robust and responsible use of synthetic tabular data.

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

# Contents

# A  Related Work

**Watermarking LLM-generated text.**  Watermarking in large language models (LLMs) can be broadly categorized into unbiased and biased approaches, both aiming to embed detectable signals without substantially degrading text quality. Unbiased watermarks preserve the original next-token distribution exactly. Aaronson (2023) draws independent pseudorandom variables and samples next token using a deterministic decoder, preserving the multinomial distribution via the Gumbel-max trick. Similarly, Kuditipudi et al. (2024) generates the next token based on inverse transform sampling of the multinomial distribution. Optimal detection rules for these two unbiased LLM watermarks are derived under the statistical framework (Li et al., 2025a). In contrast, biased watermarks perturb the token distribution to embed watermark signals. The KGW watermarking scheme (Kirchenbauer et al., 2023) randomly partitions the token vocabulary into green and red lists and then increases the sampling probability of green tokens to create detectable deviations in green-token frequency. Subsequent works have focused on improving robustness (Zhao et al., 2024) and optimizing the trade-off between detectability and text quality (Wouters, 2024; Huang and Wan, 2025). Additionally, Xie et al. (2025) and Wu et al. (2024) proposed unbiased variants of the KGW watermark by introducing decoding algorithm based on maximal coupling and reweighting strategy, respectively. However, due to their reliance on the order of tokens, these text watermarking methods can not be directly applied to structural tabular data, where attacks like row reordering are so common.

**Watermarking generated images.**  Image watermarking methods embed invisible signals either during training or sampling phase. Lukas and Kerschbaum (2023) proposes a method to watermark pre-trained GANs without access to training data. Wen et al. (2023) exploits DDIM invertibility to embed structural patterns into the frequency domain of the initial noise vector during sampling, achieving an effective and invisible image watermarking. Yang et al. (2024) implements a performance-preserving watermarking for diffusion models by incorporating diffused bit information into Gaussian noise in the latent space. Zhu et al. (2025) has explored generalizing image watermarking techniques to structural tabular data. However, these methods still either fail to support row-wise detection or suffer from poor data fidelity and limited robustness.

**Watermarking synthetic tabular data.**  Tabular watermarking methods primarily fall into two categories: sampling-phase watermarking and post-editing watermarking. The former embeds watermark into the latent space during the denoising generation phase or modifies the generative workflow. For instance, Zhu et al. (2025) implements row-wise watermark embedding using self-cloning and seeded shuffling techniques, ensuring close approximation to the standard Gaussian distribution. However, due to its reliance on large diffusion models using DDIM sampling strategy, this method is unsuitable for scenarios with limited GPU resources. Fang et al. (2025) proposes MUSE, a model-agnostic method that selects watermarked rows via a pseudorandom scoring mechanism across multiple candidates, preserving fidelity but increasing generation cost. The latter offers lightweight watermarking by modifying synthesized tabular data after generation. He et al. (2024) bins continuous feature values into predefined 'green' intervals and Zheng et al. (2024) extends post-editing watermarking to tabular datasets with mixed-type

features by selectively perturbing cells within a designated value range. While model-agnostic and computationally efficient, these approaches are either restricted by the feature type or vulnerable to noise and deletion attacks. The limitations of existing tabular watermarking methods highlight opportunities for improvement in four key dimensions: fidelity, detectability, applicability, and robustness. In this work, our proposed TAB-DRW addresses these limitations, achieving superior performance across all four dimensions.

# B  Privacy-Enhanced TAB-DRW

---

**Algorithm 3** Watermarking embedding of privacy-enhanced TAB-DRW

---
1: **Input**: Tabular data $\mathbf{X} \in \mathbb{R}^{N \times p}$, parameters $\gamma \in [0,1]$ and $\delta \in [-1,1]$, watermark key $\kappa$.
2: Shuffle $\mathbf{X}$ using key-derived permutation $P_\kappa$ to obtain $\mathbf{X}P_\kappa$.
3: Transform $\mathbf{X}P_\kappa$ using YJT and standardization (still denoted as $\mathbf{X}P_\kappa$ for simplicity).
4: **for** each row $\boldsymbol{x}$ in $\mathbf{X}P_\kappa$ **do**
5:     Compute $\boldsymbol{y} \leftarrow \text{DFT}(\boldsymbol{x})$ and generate pseudorandom bits $\{\zeta_t\}_{t=1}^m$ via Algorithm 2.
6:     Modify $\boldsymbol{y}$ according to soft variant (2) to obtain $\boldsymbol{y}^{\text{wm}}$.
7:     Recover $\boldsymbol{x}^{\text{wm}} \leftarrow \text{IDFT}(\boldsymbol{y}^{\text{wm}})$.
8: **end for**
9: Collect each $\boldsymbol{x}^{\text{wm}}$ to form a matrix $\mathbf{X}^{\text{wm}}$.
10: Apply inverse standardization and YJT to $\mathbf{X}^{\text{wm}}$, then unshuffle it to obtain $\mathbf{X}^{\text{wm}}P_\kappa^{-1}$.
11: Perform rounding and clipping if needed, and release.

---

Inspired by the KGW watermark (Kirchenbauer et al., 2023), we extend TAB-DRW with a privacy-enhanced variant designed to increase the difficulty of watermark removal. The modification is straightforward: prior to **Step 1** of watermark embedding, the columns of the tabular data are shuffled according to a pseudorandom permutation determined by the watermark key $\kappa$ (which can be shared with the key in Algorithm 2), and after **Step 3**, the columns are reshuffled back to their original order. During detection, a verifier holding the correct key can reproduce the same column permutation to obtain the watermarked frequency-domain representation. See Figure 7 and Algorithm 3 for the complete procedure of the privacy-enhanced TAB-DRW.

Privacy-enhanced TAB-DRW satisfies two crucial requirements:

1. **The key-dependent variability in the frequency-domain representation does not substantially affect watermark distortion or detectability**. In other words, given randomly selected watermark keys, the privacy-enhanced TAB-DRW should exhibit nearly consistent performance in terms of data fidelity and watermark detectability. From a theoretical perspective, since $P_\kappa$ is an orthogonal permutation matrix and the DFT/IDFT are unitary, inserting $P_\kappa$ before and $P_\kappa^{-1}$ after the frequency-domain transformation amounts to a norm-preserving change of entries in the original domain, meaning the total $\ell_2$ distortion remains unchanged. Moreover, because detection evaluates sign alignment in the same keyed frequency coordinates, the resulting $Z$-score is invariant up to index relabeling. Empirical evidence supporting this claim is provided in Table 20 of Appendix G.3.

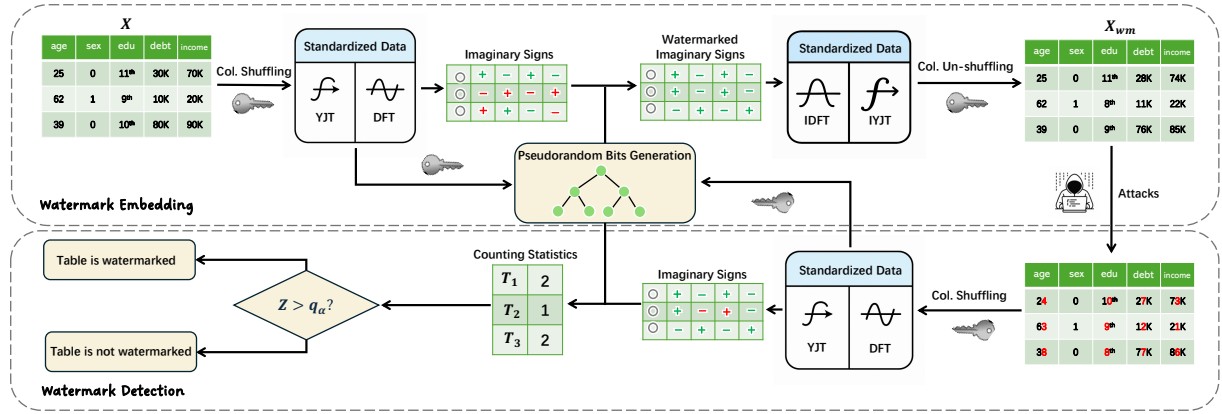

Figure 7: Work flow of privacy-enhanced TAB-DRW.

2. **The approach supports multi-key scenarios, as a watermark embedded with one key cannot be detected using another, thereby effectively avoiding false positives.** Furthermore, the collision-free key space must be sufficiently large to support large-scale deployment. The motivation behind this design lies in the sensitivity of the row-wise DFT to column order: frequency-domain representations derived from different pseudorandom permutations are nearly independent and exhibit nontrivial discrepancies. As a result, the watermark key is effectively encoded into the frequency-domain watermark signal as a unique, secret pattern. Furthermore, this sensitivity induces a combinatorially large key space of size $\mathcal{O}(p!)$ for a tabular dataset with $p$ columns. Comprehensive empirical evaluations are presented in Table 21 of Appendix G.3.

## C  Missing Details from Section 2

**A toy example.** We illustrate Algorithm 2 with the toy setting used in Figure 2: let $N = 3$, $p = 13$, and hence $m = \lfloor (p-1)/2 \rfloor = 6$. Suppose the secret key selects the column subset $\mathcal{I} = \{1, 5, 9\}$, and the sums of the three rows over $\mathcal{I}$ are $-0.8$, $0.3$, and $1.1$, respectively. Consider the second row as the target row $\boldsymbol{x}^*$. Since its sum is the second-largest among the three rows, its rank is $x^*_{\text{rank}} = 1$. After normalization in Line 6, we obtain $x^*_{\text{rank}} = 1/(3-1) = 0.5$. Because $m = 6$, the interval $[0, 1]$ is partitioned into $2^{\lceil m/2 \rceil} = 2^3 = 8$ equal-sized bins, and the tree depth is $\lceil m/2 \rceil = 3$. We then traverse the path determined by $x^*_{\text{rank}} = 0.5$. At level $j = 1$, Line 8 gives $k = \min(2^1 - 1, \lfloor 2^1 \cdot 0.5 \rfloor) = 1$, so Line 9 appends $[0, 1]$ to **S**. At level $j = 2$, we have $k = \min(2^2 - 1, \lfloor 2^2 \cdot 0.5 \rfloor) = 2$, so we append another $[0, 1]$. At level $j = 3$, we have $k = \min(2^3 - 1, \lfloor 2^3 \cdot 0.5 \rfloor) = 4$, so we append $[1, 0]$. Therefore, after the three levels, $\mathbf{S} = [0, 1, 0, 1, 1, 0]$. Since $|\mathbf{S}| = m = 6$, the truncation step does not remove any entry, and the final pseudorandom bit sequence for the target row is $[0, 1, 0, 1, 1, 0]$. Figure 3 visualizes the nodes visited in Lines 7–9 for this example.

**Connection to Gray codes.** From the perspective of Gray codes, our pseudorandom bit generation scheme can be viewed as a special case of a 2-Gray code. We present the formal construction process below.

Let $n \in \mathbb{N}$. Denote by $\{0, 1\}^n$ the set of all binary strings of length $n$, and by $d_H(\cdot, \cdot)$ the

Hamming distance on $\{0,1\}^n$. Let $G_n = (g_0^n, g_1^n, \ldots, g_{2^n-1}^n)$ with $g_i^n \in \{0,1\}^n$ be a standard $n$-bit 1-Gray code, specified up to cyclic permutation. By definition, $d_H(g_i^n, g_{i+1}^n) = 1$ for all $i \in \{0, \ldots, 2^n - 1\}$. We define the pair-encoding map $\varphi : \{0,1\} \to \{0,1\}^2, \varphi(0) = 10, \varphi(1) = 01$ and extend $\varphi$ to a map $\phi : \{0,1\}^n \to \{0,1\}^{2n}$ by applying it coordinatewise: for $g^n = b_1 b_2 \cdots b_n \in \{0,1\}^n$, $b_j \in \{0,1\}$, let $\phi(g^n) = \varphi(b_1)\varphi(b_2)\cdots\varphi(b_n)$. For $n \in \mathbb{N}$, we then can define the set $H_{2n} = (h_0^{2n}, h_1^{2n}, \ldots, h_{2^n-1}^{2n})$ and $H_{2n-1} = (\pi(h_0^{2n}), \pi(h_1^{2n}), \ldots, \pi(h_{2^n-1}^{2n}))$, where $h_i^{2n} := \phi(g_i^n) \in \{0,1\}^{2n}$ and $\pi : \{0,1\}^{2n} \to \{0,1\}^{2n-1}$ be the projection that deletes the last coordinate: $\pi(x_1, \ldots, x_{2n}) = (x_1, \ldots, x_{2n-1})$.

By construction, the family $\{H_n\}$ forms a collection of 2-Gray codes, and any two consecutive codewords $h_i^n$ and $h_{i+1}^n$ satisfy $d_H(h_i^n, h_{i+1}^n) \leq 2$. In our paper, a tree of depth $N$ stores the sequence $\{H_n\}_{n=1}^{2N}$, and each rank bin corresponds to a distinct $h_i^{2N}$ or $\pi(h_i^{2N})$. Compared with using a standard 1-Gray code, this construction has the advantage of slowing the growth rate of rank bins as the number of table columns increases, since $|H_n| = 2^{\lceil n/2 \rceil}$ whereas $|G_n| = 2^n$. This reduction leads to greater robustness of bit retrieval against rank shifts. In Appendix G.1, we provide additional evaluations on using a 1-Gray code for bit sequence generation .

## D  Missing Details from Section 3

**Soundness of robustness analysis in the transformed domain.**  This appendix provides detailed empirical evidence supporting Remark 3.

We first provide a controlled case study to quantify the impact of YJT parameter refitting. We generate multivariate Gaussian tables with row counts $N \in \{100, 1000, 10000\}$ and column counts $p \in \{10, 50, 100\}$. Two covariance structures are considered: the identity matrix and an AR(1) matrix $\Sigma_{ij} = \rho^{|i-j|}$ with $\rho = 0.4$. Each table is watermarked using TAB-DRW with $(\gamma, \delta) = (0.5, 0.5)$. The YJT parameters $\lambda$, mean $\mu$, and standard deviation $\sigma$ are recorded before and after watermarking and averaged across all columns. The results in Table 6 show that TAB-DRW introduces negligible perturbations to these parameters across different dimensions and covariance structures.res.

We also provide additional empirical evaluations by comparing detection performance under 1) **idealized setting:** No parameter refitting, as assumed in Section 3, and 2) **practical setting:** With parameter refitting, as used in experiments. As shown in Table 7, the impact of distribution shifts on detection performance is negligible relative to post-processing attacks, and the embedded watermark remains highly detectable even under such shifts. These results demonstrate both the robustness of our approach and the validity of conducting robustness analysis in the transformed domain.

**Soundness of robustness analysis under Gaussian setting.**  Real-world tabular data are highly heterogeneous and rarely follow a strict multivariate Gaussian distribution. However, after applying the Yeo–Johnson transformation (YJT), the data typically become much closer to Gaussian. As discussed earlier, YJT standardizes heterogeneous feature scales and reduces skewness, enabling more tractable analysis in the transformed space.

Although the Gaussian setting does not fully capture the complexity of real-world data, deriving closed-form robustness guarantees under arbitrary, non-Gaussian distributions is generally

Table 6: YJT parameters summary (before vs. after watermarking) across varying dimensions and covariance structures.

| $\Sigma$ | N | p | $\lambda$ Before | $\lambda$ After | $\mu$ Before | $\mu$ After | $\sigma$ Before | $\sigma$ After |
|---|---|---|---|---|---|---|---|---|
| Identity | 100 | 10 | 1.0164 | 1.0117 | -0.0230 | -0.0229 | 0.9792 | 0.9592 |
| Identity | 100 | 50 | 1.0065 | 1.0030 | -0.0017 | -0.0027 | 0.9957 | 0.9835 |
| Identity | 100 | 100 | 0.9743 | 0.9807 | -0.0190 | -0.0166 | 0.9917 | 0.9817 |
| Identity | 1000 | 10 | 1.0113 | 1.0113 | 0.0418 | 0.0420 | 1.0002 | 0.9883 |
| Identity | 1000 | 50 | 0.9971 | 0.9940 | -0.0070 | -0.0081 | 1.0020 | 0.9939 |
| Identity | 1000 | 100 | 0.9885 | 0.9861 | -0.0056 | -0.0064 | 1.0014 | 0.9943 |
| Identity | 10000 | 10 | 0.9970 | 0.9999 | 0.0017 | 0.0026 | 0.9988 | 0.9889 |
| Identity | 10000 | 50 | 1.0029 | 1.0027 | 0.0000 | -0.0001 | 1.0000 | 0.9925 |
| Identity | 10000 | 100 | 1.0003 | 1.0011 | 0.0017 | 0.0020 | 0.9997 | 0.9930 |
| AR(1) | 100 | 10 | 0.9849 | 0.9735 | 0.0075 | 0.0028 | 1.0367 | 1.0231 |
| AR(1) | 100 | 50 | 1.0158 | 1.0128 | 0.0072 | 0.0064 | 0.9766 | 0.9680 |
| AR(1) | 100 | 100 | 0.9917 | 0.9921 | -0.0222 | -0.0223 | 0.9811 | 0.9724 |
| AR(1) | 1000 | 10 | 0.9866 | 0.9892 | -0.0024 | -0.0015 | 0.9963 | 0.9873 |
| AR(1) | 1000 | 50 | 0.9926 | 0.9911 | -0.0046 | -0.0051 | 0.9988 | 0.9926 |
| AR(1) | 1000 | 100 | 0.9972 | 0.9967 | 0.0010 | 0.0009 | 0.9954 | 0.9898 |
| AR(1) | 10000 | 10 | 1.0037 | 1.0051 | 0.0008 | 0.0013 | 0.9958 | 0.9873 |
| AR(1) | 10000 | 50 | 1.0023 | 1.0032 | -0.0051 | -0.0048 | 0.9998 | 0.9938 |
| AR(1) | 10000 | 100 | 0.9998 | 0.9998 | -0.0001 | -0.0000 | 0.9999 | 0.9946 |

Table 7: Detection performance under different parameter-refitting settings. Each entry reports the average Z-score over 1K rows, evaluated using TAB-DRW with $(\gamma, \delta) = (0.5, 0.5)$ under 100 trials.

| Dataset | Idealized setting | Practical setting |
|---|---|---|
| Adult | 15.13±1.04 | 12.81±1.17 |
| Magic | 30.47±0.85 | 27.34±0.93 |
| Shoppers | 21.14±0.84 | 18.18±1.28 |
| Default | 19.02±0.85 | 15.98±0.92 |
| Drybean | 41.36±0.98 | 38.03±1.03 |

intractable. Our aim is not to provide universal theoretical guarantees, but to clarify the underlying robustness mechanism of our method. In particular, we show how sign alignment in the frequency domain, together with the hyperparameters $(\gamma, \delta)$, preserves the watermark signal under perturbations. Therefore, we adopt the multivariate Gaussian model as a simplified yet widely used analytical tool to make this intuition concrete. As the saying goes, "All models are wrong, but some are useful." Our analysis is intended to shed light on why our method is robust—not to claim robustness under all possible data distributions.

# E    Theoretical Analysis and Proofs

## E.1    Proof of Proposition 1

*Proof of Proposition 1.* Let $\Delta y_{i,j} = y_{i,j}^{\text{wm}} - y_{i,j}$ denote the entry-wise difference in the frequency domain, then by Def. 2 and the Algorithm 1 with soft parameters $(\gamma, \delta)$, we have

$$
\Delta y_{i,k} = \begin{cases} -\mathtt{i}(1 + \delta) \cdot \Im(y_{i,k}), & k \in S, \\ \mathtt{i}(1 + \delta) \cdot \Im(y_{i,p-k}), & p - k \in S, \\ 0, & \text{otherwise.} \end{cases}
$$

By the inverse DFT as defined in Def. 2, the entry-wise difference $\Delta x_{i,j}$ is given by

$$
\begin{aligned}
\Delta x_{i,j} &= \frac{1}{\sqrt{p}} \left[ \sum_{k \in S} \Delta y_{i,k} e^{\mathtt{i}\frac{2\pi}{p} kj} + \sum_{p-k \in S} \Delta y_{i,k} e^{\mathtt{i}\frac{2\pi}{p} kj} \right] \\
&= \frac{2(1 + \delta)}{\sqrt{p}} \left[ \sum_{k \in S} \Im(y_{i,k}) \sin \frac{2\pi}{p} kj \right].
\end{aligned}
$$

Plugging in $\Im(y_{i,k}) = -\frac{1}{\sqrt{p}} \sum_{n=0}^{p-1} x_{in} \sin \frac{2\pi kn}{p}$ leads to

$$
\Delta x_{i,j} = -\frac{2(1 + \delta)}{p} \sum_{n=0}^{p-1} \left[ \sum_{k \in S} \sin \frac{2\pi kn}{p} \sin \frac{2\pi kj}{p} \right] x_{in},
$$

which is precisely $\Delta x_{i,j} = -\alpha \, \boldsymbol{\beta}_j^\top \boldsymbol{x}_i$ with $\alpha = \frac{2(1+\delta)}{p}$ and the stated $\boldsymbol{\beta}_j$ and $\boldsymbol{x}_i$. $\qquad \square$

## E.2    Proof of Theorem 1

We prove items one by one.

1. **Mean.** For each column $j = 0, \ldots, p - 1$, we have

$$
\frac{1}{N} \sum_{i=1}^N \Delta x_{i,j} = -\alpha \, \boldsymbol{\beta}_j^\top \left( \frac{1}{N} \sum_{i=1}^N \boldsymbol{x}_i \right) = 0,
$$

since each column is centered.

2. **Pearson correlation coefficients (PCC).** Given that each column is centered and standardized, the difference of PCC between each column pair $(j, \ell)$ is given by

$$
\Delta r_{j\ell} = \frac{1}{N} \sum_{i=1}^N (x_{i,j} \Delta x_{i,\ell} + x_{i,\ell} \Delta x_{i,j} + \Delta x_{i,j} \Delta x_{i,\ell}).
$$

Plugging $\Delta x_{i,j} = -\alpha \, \boldsymbol{\beta}_j^\top \boldsymbol{x}_i$ leads to

$$
\Delta r_{j\ell} = -\alpha \left( [\boldsymbol{\Sigma} \boldsymbol{\beta}_\ell]_j + [\boldsymbol{\Sigma} \boldsymbol{\beta}_j]_\ell \right) + \alpha^2 \, \boldsymbol{\beta}_j^\top \boldsymbol{\Sigma} \boldsymbol{\beta}_\ell,
$$

where $\boldsymbol{\Sigma} = \frac{1}{N} \mathbf{X}^\top \mathbf{X}$ with $\text{diag}(\boldsymbol{\Sigma}) = \mathbb{I}$.

3. **Empirical distribution.** Consider the coupling matching $x_{i,j}$ to $x_{i,j} + \Delta x_{i,j}$ for each $i$, we bound the transport cost as below:

$$\mathcal{W}_2^2(\rho_j, \nu_j) \leq \frac{1}{N} \sum_{i=1}^{N} (\Delta x_{i,j})^2 = \alpha^2 \boldsymbol{\beta}_j^\top \boldsymbol{\Sigma} \boldsymbol{\beta}_j,$$

which leads to the claimed inequality.

### E.3  Proof of Theorem 2

**Lemma 1** (Gaussian tail bound). *Let $\Phi(u)$ denote the standard normal CDF and $Q(u) := 1 - \Phi(u)$. For any $u > 0$,*

$$Q(u) \leq \frac{1}{2} e^{-u^2/2}.$$

*Proof of Lemma 1.* We discuss the bound under two cases.
**Case 1**: When $u > \sqrt{\frac{2}{\pi}}$, through integration by parts, we have

$$Q(u) = \frac{1}{\sqrt{2\pi}} \int_u^\infty e^{-t^2/2} dt \leq \frac{1}{\sqrt{2\pi}} \left[ \frac{e^{-u^2/2}}{u} - \int_u^\infty \frac{e^{-t^2/2}}{t^2} dt \right].$$

Dropping the negative integral preserves the inequality, yielding

$$Q(u) \leq \frac{1}{u\sqrt{2\pi}} e^{-u^2/2} \leq \frac{1}{2} e^{-u^2/2}.$$

**Case 2**: When $0 < u \leq \sqrt{\frac{2}{\pi}}$, we have

$$\frac{d}{du}\left( \frac{1}{2} e^{-u^2/2} \right) = -\frac{u}{2} e^{-u^2/2} \geq -\frac{1}{\sqrt{2\pi}} e^{-u^2/2} = \frac{d}{du} Q(u),$$

where the inequality follows from $u \leq \sqrt{\frac{2}{\pi}}$. Integrating from 0 to $u$ gives

$$\int_0^u d\left( \frac{1}{2} e^{-t^2/2} \right) \geq \int_0^u dQ(t) \Rightarrow Q(u) \leq \frac{1}{2} e^{-u^2/2}.$$

Combining the two cases yields the stated bound.  $\square$

**Lemma 2** (Noise in the frequency domain). *Let $\boldsymbol{\varepsilon} = (\varepsilon_0, \ldots, \varepsilon_{p-1})^\top \sim \mathcal{N}(\mathbf{0}, \sigma^2 \mathbf{I}_p)$ be a real-valued Gaussian noise vector of length $p$. Apply the DFT in Def. 2 to obtain $\hat{\boldsymbol{\varepsilon}} = (\hat{\varepsilon}_0, \ldots, \hat{\varepsilon}_{p-1})^\top$. Denote by*

$$z_t = \Im(\hat{\varepsilon}_t), \quad t = 1, \ldots, m,$$

*the imaginary part of the noise component at the $t$-th effective entry. Then*

$$z_t \sim \mathcal{N}\left( 0, \frac{\sigma^2}{2} \right).$$

*Proof of Lemma 2.* Denote $\theta_{t,n} := \frac{2\pi t n}{p}$, we have

$$z_t = -\frac{1}{\sqrt{p}} \sum_{n=0}^{p-1} \varepsilon_n \sin\left(\theta_{t,n}\right).$$

Note that $z_t$ is a linear combination of independent Gaussian variables, hence still be a Gaussian with zero mean. Since $\mathrm{Var}(\varepsilon_n) = \sigma^2$ and the $\varepsilon_n$'s are independent,

$$\mathrm{Var}[z_t] = \frac{1}{p}\sigma^2 \sum_{n=0}^{p-1} \sin^2\left(\theta_{t,n}\right).$$

Using the trigonometric identity $\sin^2 u = \frac{1}{2}\left(1 - \cos 2u\right)$,

$$\sum_{n=0}^{p-1} \sin^2(\theta_{t,n}) = \frac{p}{2} - \frac{1}{2}\sum_{n=0}^{p-1} \cos(2\theta_{t,n}).$$

The second sum is a geometric series whose value is 0 whenever $t \notin \{0, \frac{p}{2}\}$:

$$\sum_{n=0}^{p-1} e^{i\frac{4\pi t n}{p}} = \frac{1 - e^{i4\pi t}}{1 - e^{i\frac{4\pi t}{p}}} = 0.$$

Hence $\sum_{n=0}^{p-1} \sin^2(\theta_{t,n}) = \frac{p}{2}$ for each $t = 1, \ldots, m$. Substituting back,

$$\mathrm{Var}[z_t] = \frac{\sigma^2}{p} \cdot \frac{p}{2} = \frac{\sigma^2}{2}.$$

$\square$

**Lemma 3** (Standard Z-score). *If pseudorandom bits $\zeta_{i,j} \overset{\text{i.i.d.}}{\sim} Bernoulli(0.5)$ and are independent of effective entries $\{y_{i,j}\}$, then $\{T_i\}_{i=1}^{N}$, as defined in Section 2, i.i.d. follows $B(m, \frac{1}{2})$ under $H_0$, thus has expected value $\frac{m}{2}$ and variance $\frac{m}{4}$. By Central Limit Theorem, the Z-score under $H_0$ follows*

$$Z = \frac{\sum_{i=1}^{N} T_i - \frac{mN}{2}}{\sqrt{\frac{mN}{4}}} \overset{d}{\to} \mathcal{N}(0,1) \quad as \quad N \to \infty.$$

*Proof of Lemma 3.* For each index pair $(i, j)$ of effective entries, define the events

$$E_{i,j} := \{\mathrm{sign}(\Im(y_{i,j})) = 2\,\zeta_{i,j} - 1\}, \qquad A_{i,j} := \{\mathrm{sign}(\Im(y_{i,j})) = 1\}.$$

We will show that the indicator variables $\{\mathbb{I}(E_{i,j})\}_{i,j}$ are independent and identically distributed as Bernoulli(0.5).

First, set

$$p_{i,j} := \mathbb{P}\left(\mathrm{sign}(\Im(y_{i,j})) = 1\right).$$

By conditioning on $\zeta_{i,j} \in \{0,1\}$ and using the fact that $\mathbb{P}(\zeta_{i,j} = 1) = \mathbb{P}(\zeta_{i,j} = 0) = \frac{1}{2}$, we obtain

$$\mathbb{P}(E_{i,j}) = p_{i,j} \, \mathbb{P}(\zeta_{i,j} = 1) + (1 - p_{i,j}) \, \mathbb{P}(\zeta_{i,j} = 0) = \frac{1}{2} \, p_{i,j} + \frac{1}{2} \, (1 - p_{i,j}) = \frac{1}{2}.$$

Hence each $\mathbb{I}(E_{i,j}) \sim \mathrm{Bernoulli}(0.5)$.

To verify independence, for any finite index set $\mathcal{I} \subseteq \{(i,j) \colon 1 \le i \le N, 1 \le j \le m\}$. we consider a family of events

$$\mathcal{B} = \left\{ B_{\mathcal{I}_1, \mathcal{I}_2} \colon \mathcal{I}_1 \cup \mathcal{I}_2 = \mathcal{I}, \mathcal{I}_1 \cap \mathcal{I}_2 = \emptyset \right\}, \quad B_{\mathcal{I}_1, \mathcal{I}_2} = \left( \bigcap_{(i,j) \in \mathcal{I}_1} A_{i,j} \right) \cap \left( \bigcap_{(i,j) \in \mathcal{I}_2} A_{i,j}^c \right).$$

Clearly $\mathcal{B}$ is a partition of the sample space $\Omega$, hence we have

$$\mathbb{P}\left( \bigcap_{(i,j) \in \mathcal{I}} E_{i,j} \right) = \sum_{B_{\mathcal{I}_1, \mathcal{I}_2} \in \mathcal{B}} \mathbb{P}(B_{\mathcal{I}_1, \mathcal{I}_2}) \prod_{(i,j) \in \mathcal{I}_1} \mathbb{P}(\zeta_{i,j} = 1) \prod_{(i,j) \in \mathcal{I}_2} \mathbb{P}(\zeta_{i,j} = 0)$$
$$= \sum_{B_{\mathcal{I}_1, \mathcal{I}_2} \in \mathcal{B}} \mathbb{P}(B_{\mathcal{I}_1, \mathcal{I}_2}) \frac{1}{2^{|\mathcal{I}|}} = \frac{1}{2^{|\mathcal{I}|}} = \prod_{(i,j) \in \mathcal{I}} \mathbb{P}(E_{i,j}),$$

This implies that the collection of events $\{E_{i,j}\}_{i,j}$ is mutually independent. Together with the fact that $\mathbb{P}(E_{i,j}) = \frac{1}{2}$, this completes the proof.

$\square$

*Proof of Theorem 2.* We continue with the notations established in Lemmas 2 and 3. Let

$$\mathbf{X} = \{x_{i,j}\} \in \mathbb{R}^{N \times p}, \quad \boldsymbol{x}_i := (x_{i,0}, \ldots, x_{i,p-1}) \overset{\text{i.i.d.}}{\sim} \mathcal{N}(0, \boldsymbol{\Sigma}),$$

and denote the frequency-domain representation by $\mathbf{Y} \in \mathbb{C}^{N \times p}$. Then for each row, the effective entries satisfy:

$$\Im(y_t) = -\frac{1}{\sqrt{p}} \sum_{n=0}^{p-1} x_n \sin(\theta_{t,n}),$$

where $\theta_{t,n} = \frac{2\pi t n}{p}$. Let $\boldsymbol{s}_t$ denotes $(\sin(\theta_{t,0}), \ldots, \sin(\theta_{t,p-1})) \in \mathbb{R}^{1 \times p}$. Since each $\boldsymbol{x}_i$ is Gaussian, $\Im(y_t)$ is Gaussian with zero mean and

$$\mathrm{Var}\left[ \Im(y_t) \right] = \frac{1}{p} \boldsymbol{s}_t^\top \boldsymbol{\Sigma} \boldsymbol{s}_t \in \left[ \frac{\lambda_{\min}}{p} \|\boldsymbol{s}_t\|_2^2, \frac{\lambda_{\max}}{p} \|\boldsymbol{s}_t\|_2^2 \right] = \left[ \frac{\lambda_{\min}}{2}, \frac{\lambda_{\max}}{2} \right],$$

where $\|\boldsymbol{s}_t\|_2^2 = \frac{p}{2}$ follows from Lemma 2 and $\lambda_{\min}$ and $\lambda_{\max}$ denote the smallest and largest eigenvalue of the covariance matrix $\boldsymbol{\Sigma}$, respectively.

Given a pseudorandom bit $\zeta_t \in \{0,1\}$, the process of watermark embedding in Algorithm 1 replaces $\Im(y_t)$ by

$$\Im\left(y_t^{\mathrm{wm}}\right) = \begin{cases} -\delta \cdot \Im(y_t), & \text{if } \Im(y_t) \cdot (2\zeta_t - 1) < 0 \text{ and } |\Im(y_t)| \le \mathrm{Quantile}_\gamma(\{|\Im(y_t)|\}_{t=1}^m), \\ \Im(y_t), & \text{otherwise,} \end{cases}$$

Let

$$\alpha_t := |\Im(y_t^{\text{wm}})|, \quad \frac{\lambda}{2} := \text{Var}[\Im(y_t)] \in \left[\frac{\lambda_{\min}}{2}, \frac{\lambda_{\max}}{2}\right],$$

and define

$$F(x) := \frac{2}{\sqrt{\pi\lambda}} e^{-\frac{x^2}{\lambda}} \mathbb{I}(x \geq 0), \quad F_\delta(x) := \frac{2}{\delta\sqrt{\pi\lambda}} \exp\left(-\frac{x^2}{\delta^2\lambda}\right) \mathbb{I}(x \geq 0),$$

which are the PDFs of $\alpha_t$ and $\delta\,\alpha_t$, respectively. Under large $p$, there are three scenarios for each $t$:

- **Case 1**: With probability $\frac{1}{2}$, $\alpha_t \sim F$ and $\Im(y_t^{\text{wm}}) \cdot (2\zeta_t - 1) > 0$.

- **Case 2**: With probability $\frac{\gamma}{2}$, $\alpha_t \sim F_\delta$ and $\Im(y_t^{\text{wm}}) \cdot (2\zeta_t - 1) > 0$.

- **Case 3**: With probability $\frac{1-\gamma}{2}$, $\alpha_t \sim F$ and $\Im(y_t^{\text{wm}}) \cdot (2\zeta_t - 1) < 0$.

**Sign-flip probability under additive noise.** Let $z_t \sim \mathcal{N}(0, \frac{\sigma^2}{2})$ be the imaginary-part noise as derived in Lemma 2. Conditioned on $\alpha_t = x$, the probability that noise flips the sign of $\Im(y_t^{\text{wm}})$ in **Case 1** and **Case 3** is

$$\begin{aligned}
\mathbb{P}_{\text{flip}}(\sigma) &= \mathbb{P}\left(z_t > \alpha_t | \alpha_t \sim F\right) \\
&= \int_0^\infty \mathbb{P}\left(z_t > \alpha_t | \alpha_t = x\right) F(x) dx \\
&= \int_0^\infty \frac{2}{\sqrt{\pi\lambda}} e^{-\frac{x^2}{\lambda}} Q\left(\frac{\sqrt{2}\,x}{\sigma}\right) dx \\
&\leq \frac{1}{\sqrt{\pi\,\lambda_{\min}}} \int_0^{\sqrt{\frac{\lambda_{\min}}{2}}} e^{-x^2\left(\frac{1}{\lambda_{\min}} + \frac{1}{\sigma^2}\right)} dx \\
&\quad + \frac{1}{\sqrt{\pi\,\lambda_{\max}}} \int_{\sqrt{\frac{\lambda_{\max}}{2}}}^\infty e^{-x^2\left(\frac{1}{\lambda_{\max}} + \frac{1}{\sigma^2}\right)} dx \\
&\quad + \frac{e^{-\frac{1}{2}}}{\sqrt{2\pi}} \int_{\sqrt{\frac{\lambda_{\min}}{2}}}^{\sqrt{\frac{\lambda_{\max}}{2}}} \frac{e^{-\frac{x^2}{\sigma^2}}}{x} dx \\
&= \frac{\sigma}{\sqrt{\sigma^2 + \lambda_{\min}}} \left[\Phi\left(\sqrt{1 + \frac{\lambda_{\min}}{\sigma^2}}\right) - \frac{1}{2}\right] \\
&\quad + \frac{\sigma}{\sqrt{\sigma^2 + \lambda_{\max}}} \left[1 - \Phi\left(\sqrt{1 + \frac{\lambda_{\max}}{\sigma^2}}\right)\right] \\
&\quad + \frac{1}{\sqrt{8\pi e}} \left[E_1\left(\frac{\lambda_{\min}}{2\sigma^2}\right) - E_1\left(\frac{\lambda_{\max}}{2\sigma^2}\right)\right],
\end{aligned}$$

where the inequality follows from Lemma 1 and the local monotonicity of $F(x)$. Similarly, if the amplitude of $\alpha_t$ is scaled by $\delta$, we obtains $\mathbb{P}_{\text{flip}}^{(\delta)}(\sigma) = \mathbb{P}\left(z_t > \alpha_t | \alpha_t \sim F_\delta\right) \leq \mathcal{I}(\frac{\sigma}{\delta})$, where

$$\mathcal{I}(s) := \frac{s}{\sqrt{s^2 + \lambda_{\min}}}\left[\Phi\left(\sqrt{1 + \frac{\lambda_{\min}}{s^2}}\right) - \frac{1}{2}\right] + \frac{s}{\sqrt{s^2 + \lambda_{\max}}}\left[1 - \Phi\left(\sqrt{1 + \frac{\lambda_{\max}}{s^2}}\right)\right] + \frac{1}{\sqrt{8\pi e}}\left[E_1\left(\frac{\lambda_{\min}}{2s^2}\right) - E_1\left(\frac{\lambda_{\max}}{2s^2}\right)\right].$$

**Alignment probability after attack.** Let $p_{i,j}$ be the probability that the $j$-th effective entry in row $i$ maintains alignment with its corresponding pseudorandom bit under attack. Combining

the three cases above, we have

$$p_{i,j} = \frac{1}{2}\left(1 - \mathbb{P}_{\text{flip}}(\sigma)\right) + \frac{\gamma}{2}\left(1 - \mathbb{P}_{\text{flip}}^{(\delta)}(\sigma)\right) + \frac{1-\gamma}{2}\mathbb{P}_{\text{flip}}(\sigma)$$

$$= \frac{1+\gamma}{2} - \frac{\gamma}{2}\left[\mathbb{P}_{\text{flip}}(\sigma) + \mathbb{P}_{\text{flip}}^{(\delta)}(\sigma)\right] \tag{6}$$

$$\geq \frac{1+\gamma}{2} - \frac{\gamma}{2}\left[\mathcal{I}(\sigma) + \mathcal{I}\left(\frac{\sigma}{\delta}\right)\right]$$

**Lower bound on the expected Z-score.** Under this setting, we recall Lemma 3 for the standard Z-score $Z = \frac{\sum_{i,j}\mathbb{I}\{E_{i,j}\} - \frac{mN}{2}}{\sqrt{\frac{mN}{4}}}$, then we obtain

$$\mathbb{E}\left[Z(\gamma,\delta,\sigma)\right] = \frac{mN\, p_{i,j} - \frac{mN}{2}}{\sqrt{\frac{mN}{4}}} \geq \sqrt{mN}\gamma\left[1 - \mathcal{I}(\sigma) - \mathcal{I}(\frac{\sigma}{\delta})\right],$$

which completes the proof. $\qquad\square$

### E.4  Proof of Corollary 1

*Proof.* Denote the random variables after noise perturbation as below: $I_{i,j} := \mathbb{I}\{\Im(y_{i,j})(2\zeta_{i,j}-1) > 0\}$, $T_i := \sum_{j=1}^{m} I_{i,j}$, and $S_N := \sum_{i=1}^{N} T_i$. We omit their explicit dependence on hyperparameters $(\gamma,\delta,\sigma)$ here for simplicity. From Lemma 3, the Z-score follows

$$Z = \frac{S_N - \frac{mN}{2}}{\sqrt{\frac{mN}{4}}}.$$

By Eq.(6), we have

$$\mathbb{E}_{H_1}[S_N] = \sum_{i=1}^{N}\sum_{j=1}^{m} p_{i,j} \geq mN\left(\frac{1+\gamma}{2} - \frac{\gamma}{2}\left[\mathcal{I}(\sigma) + \mathcal{I}(\frac{\sigma}{\delta})\right]\right).$$

Then for a one-sided level-$\alpha$ test with threshold $q_\alpha$, we have

$$\{Z \leq q_\alpha\} \subseteq \{S_N - \mathbb{E}_{H_1}[S_N] \leq -t_N\}, \quad t_N := \left(\sqrt{mN}\gamma\left[1 - \mathcal{I}(\sigma) - \mathcal{I}(\frac{\sigma}{\delta})\right] - q_\alpha\right)\frac{\sqrt{mN}}{2},$$

Since $T_i$ are independent and bounded in $[0, m]$ (i.i.d. rows and row-wise watermarking), we apply Hoeffding's inequality:

$$\mathbb{P}_{H_1}\{Z \leq q_\alpha\} \leq \exp\left(-\frac{2t_N^2}{Nm^2}\right) = \exp\left(-\frac{\left(\sqrt{mN}\gamma\left[1 - \mathcal{I}(\sigma) - \mathcal{I}(\frac{\sigma}{\delta})\right] - q_\alpha\right)^2}{2m}\right).$$

Imposing $\mathbb{P}_{H_1}\{Z \leq q_\alpha\} \leq \beta$ gives a lower bound:

$$\sqrt{mN}\gamma\left[1 - \mathcal{I}(\sigma) - \mathcal{I}\left(\frac{\sigma}{\delta}\right)\right] \geq q_\alpha + \sqrt{2m\ln(1/\beta)},$$

which implies the nonasymptotic sample-size lower bound to achieve a test of power $1 - \beta$ at level $\alpha$:

$$N_{\alpha,\beta}(\gamma, \delta, \sigma) \geq \frac{\left[q_\alpha + \sqrt{2m \ln\left(\frac{1}{\beta}\right)}\right]^2}{m\gamma^2 \left[1 - \mathcal{I}(\sigma) - \mathcal{I}(\frac{\sigma}{\delta})\right]^2}.$$

$\square$

## E.5   Proof of Theorem 3

*Proof of Theorem 3.* Let $\boldsymbol{x} = (x_0, \ldots, x_{p-1})$ be a standardized row and $\boldsymbol{y} = \mathtt{DFT}(\boldsymbol{x})$. For the $t$-th effective frequency, we have

$$\Im(y_t) = -\frac{1}{\sqrt{p}} \sum_{n=0}^{p-1} x_n \sin\left(\frac{2\pi t n}{p}\right) = -\frac{1}{\sqrt{p}} \boldsymbol{s_t}^\top \boldsymbol{x}, \qquad \|\boldsymbol{s_t}\|_2^2 = \sum_{n=0}^{p-1} \sin^2\left(\frac{2\pi t n}{p}\right) = \frac{p}{2}.$$

By the $\boldsymbol{\Sigma}$–sub-Gaussian setting and linearity, $\Im(y_t)$ is sub-Gaussian with

$$v_t := \mathrm{Var}[\Im(y_t)] = \frac{1}{p} \boldsymbol{s_t}^\top \boldsymbol{\Sigma} \boldsymbol{s_t} \in \left[\frac{\lambda_{\min}}{2}, \frac{\lambda_{\max}}{2}\right], \qquad \|\Im(y_t)\|_{\psi_2} = \left\|-\frac{1}{\sqrt{p}} \boldsymbol{s_t}^\top \boldsymbol{x}\right\|_{\psi_2} \leq \kappa \sqrt{v_t}.$$

Let $\alpha_t := |\Im(y_t^{\mathrm{wm}})|$ and $z_t := \Im(\widehat{\varepsilon}_t) \sim N(0, \sigma^2/2)$ be the imaginary-part noise (Lemma 2). Following the analysis in Theorem 2, for each effective entry the alignment probability with its bit satisfies

$$p_{i,j} = \frac{1+\gamma}{2} - \frac{\gamma}{2}\left(\mathbb{P}_{\mathrm{flip}}(\sigma) + \mathbb{P}_{\mathrm{flip}}^{(\delta)}(\sigma)\right), \tag{7}$$

where $\mathbb{P}_{\mathrm{flip}}(\sigma) = \mathbb{E}\left[\mathbb{I}\{z_t > \alpha_t\}\right] = \mathbb{E}[Q(\sqrt{2}\,\alpha_t/\sigma)]$, and $\mathbb{P}_{\mathrm{flip}}^{(\delta)}$ is the same quantity with the amplitude scaled by $|\delta|$. Here $Q(u) = 1 - \Phi(u)$.

By Lemma 1, $Q(u) \leq \frac{1}{2}e^{-u^2/2}$, hence $\mathbb{P}_{\mathrm{flip}}(\sigma) \leq \frac{1}{2}\mathbb{E}\exp(-\alpha_t^2/\sigma^2)$. Write $U := \Im(y_t)$ and $Y := U^2$. For any $\theta \in (0, 1)$, Paley–Zygmund inequality gives

$$\mathbb{P}(Y \geq \theta \,\mathbb{E}Y) \geq \frac{(1-\theta)^2 \,(\mathbb{E}Y)^2}{\mathbb{E}Y^2} = \frac{(1-\theta)^2 \, v_t^2}{\mathbb{E}U^4} \geq \frac{(1-\theta)^2}{C_4 \,\kappa^4} =: \eta,$$

where we used the sub-Gaussian fourth-moment bound $\mathbb{E}U^4 \leq C_4 \,\kappa^4 v_t^2$. Thus, by conditioning on the event $\{Y \geq \theta v_t\}$,

$$\mathbb{E}\exp(-\alpha_t^2/\sigma^2) \leq (1-\eta) \cdot 1 + \eta \cdot \exp\left(-\frac{\theta v_t}{\sigma^2}\right) \leq 1 - \eta\left(1 - e^{-\theta\lambda_{\min}/(2\sigma^2)}\right).$$

Consequently,

$$\mathbb{P}_{\mathrm{flip}}(\sigma) \leq \frac{1}{2}\left[1 - \eta\left(1 - e^{-\theta\lambda_{\min}/(2\sigma^2)}\right)\right], \qquad \mathbb{P}_{\mathrm{flip}}^{(\delta)}(\sigma) \leq \frac{1}{2}\left[1 - \eta\left(1 - e^{-\theta\lambda_{\min}\delta^2/(2\sigma^2)}\right)\right].$$

Plugging these bounds into Eq.(7), then we obtain

$$p_{i,j} \geq \frac{1}{2} + \frac{\gamma\eta}{4}\left[2 - e^{-\theta\lambda_{\min}/(2\sigma^2)} - e^{-\theta\lambda_{\min}\delta^2/(2\sigma^2)}\right].$$

Using $\mathbb{E}Z = 2\sqrt{mN}\,(p_{i,j} - \frac{1}{2})$, we arrive at

$$\mathbb{E}[Z(\gamma, \delta, \sigma)] \;\geq\; \sqrt{mN}\,\gamma\,\frac{\eta}{2}\left[2 - e^{-\theta\lambda_{\min}/(2\sigma^2)} - e^{-\theta\lambda_{\min}\delta^2/(2\sigma^2)}\right].$$

Recalling that $\eta = (1-\theta)^2/(C_4\kappa^4)$ gives the result for $\rho(\kappa, \theta) = (1-\theta)^2/(2C_4\kappa^4)$. Since $\theta \in (0, 1)$ is a fixed hyperparameter, it can be tuned to obtain the tightest possible lower bound

$$\mathbb{E}\big[Z(\gamma, \delta, \sigma)\big] \;\geq\; \sqrt{mN}\,\gamma\,\sup_{\theta\in(0,1)}\left\{\rho(\kappa, \theta)\left[2 - \exp\Big(-\frac{\theta\,\lambda_{\min}}{2\,\sigma^2}\Big) - \exp\Big(-\frac{\theta\,\lambda_{\min}\,\delta^2}{2\,\sigma^2}\Big)\right]\right\}.$$

$\square$

## F  Experimental Details

### F.1  Datasets

The datasets used for evaluation are described in Table 8, where # Rows, # Categorical, # Numerical, # Continuous indicate the number of rows, the number of categorical columns, the number of numerical columns, and the number of numerical columns with continuous density function, respectively. # Train and # Test indicate the number of samples in the training and testing set for downstream machine learning tasks, respectively. The **Adult** (Becker and Kohavi, 1996) dataset was extracted from the 1994 Census database, containing 9 categorical and 6 numerical columns. The **Magic** (Bock, 2004) dataset simulates registration of high energy gamma particles in a ground-based atmospheric Cherenkov gamma telescope and consists of one categorical column and 10 numerical columns. The **Shoppers** (Sakar and Kastro, 2018) dataset quantifies online shoppers' purchasing intentions with 10 categorical columns and 8 numerical columns. The **Default** (Yeh, 2009) dataset presents the default payments of credit card clients, including 10 categorical columns and 14 numerical columns. The **Drybean** (Koklu and Özkan, 2020) dataset provides image information of seven different registered dry beans, comprising one categorical column and 16 numerical columns.

Here, we explicitly distinguish between continuous from integer-valued (discrete) numerical features, rather than conflating "numerical" with "continuous". As shown in Table 8, the **Adult** and **Default** datasets contain only discrete feature (0 continuous features), while **Magic** and **Drybean** are dominated by continuous features. Therefore, we believe that the selected benchmark datasets provide a sufficiently balanced evaluation across both discrete and continuous data types, supporting the claim that TAB-DRW is applicable to heterogeneous tabular data.

Table 8: Descriptions of datasets used in evaluation.

| Name | Domain | # Rows | # Categorical | # Numerical | # Continuous | # Train | # Test | Task |
|------|--------|--------|---------------|-------------|--------------|---------|--------|------|
| Adult | Society | 48,842 | 9 | 6 | 0 | 32,561 | 16,281 | Classification |
| Magic | Physics | 19,019 | 1 | 10 | 10 | 17,117 | 1,902 | Classification |
| Shoppers | Business | 12,330 | 8 | 10 | 3 | 11,097 | 1,233 | Classification |
| Default | Finance | 30,000 | 10 | 14 | 0 | 27,000 | 3,000 | Classification |
| Drybean | Biology | 13,611 | 1 | 16 | 14 | 12,249 | 1,362 | Classification |

## F.2 Metrics

We detail our data fidelity metrics below:

1. **Density** measures the distributional similarity between synthetic and real data. For each numerical column, we compute the Kolmogorov–Smirnov statistic (KST), for each categorical column, we compute the total variation distance (TVD). The per-column scores are then averaged to yield the overall Density score. Higher values indicate closer alignment of marginal distributions.

2. **Corr** evaluates preservation of inter-column relationships. We compute the Pearson correlation coefficient for every pair of columns and report their mean as the Corr score. Larger values indicate more faithful reproduction of real feature dependencies.

3. **C2ST** quantifies statistical indistinguishability between synthetic and real data. A logistic regression model is trained and evaluated on the training and validation sets, both of which contain a mix of real and synthetic data. We then report the complement of the ROC AUC averaged over all validation splits. Higher values indicate that the model cannot distinguish synthetic from real data.

4. **MLE**: assesses downstream machine learning utility on supervised tasks. We train an XGBoost model (Chen and Guestrin, 2016) on synthetic data, then evaluate it on the real testing set, using AUC for the classification task and RMSE for the regression task. Higher MLE scores reflect better machine learning utility of the synthetic data.

Regarding the metric for watermark detectability, we introduce the one-sided **Z-score** defined in (3) and **FPR / TPR**. For TAB-DRW, GLW, MUSE, and TabWak*, we first define a method-specific row-level statistic $T_i$, estimate its null mean and standard deviation from unwatermarked tables, and then standardize the table-level average into the unified one-sided Z-score. Specifically, $T_i$ is the sign-alignment count for TAB-DRW, the green-interval hit count for GLW, the Bernoulli row score induced by percentile-based keying for MUSE, and the valid-bit accuracy after DDIM inversion for TabWak*. For TabularMark, we instead retain the detector of Zheng et al. (2024), which is tailored to its cell-based watermarking on a single numerical attribute with 10% key-cell selection. A larger Z-score indicates stronger alignment with the embedded signal and hence better watermark detectability. To calibrate the null distribution for TAB-DRW, we synthesize 100 unwatermarked tables with 1K rows (100K rows in total) and perform Monte Carlo simulation under $H_0$. Specifically, for the estimation of an empirical critical value such as $q_{0.001}$, we conduct the following procedure:

1. Generate 100 unwatermarked tables with 1K rows (100K rows in total) using TabSyn.

2. Bootstrap sampling rows from 100K rows to construct 100K synthetic tables for watermark detection.

3. Set the 100-th order-statistic $Z_{(100)}$ as the threshold.

Then we have $F_{H_0}(Z_{(100)}) \sim \text{Beta}(100, 99901)$. By Clopper-Pearson interval, the estimation procedure above is sufficient to calibrate the critical value $q_{0.001}$, since the realized FPR has a 95% confidence interval of roughly $0.001 \pm 2 \times 10^{-4}$.

Table 9 reports the empirical mean and standard deviation of the Tab-Drw row-level statistic $T_i$ under $H_0$, together with the corresponding critical values $q_\alpha$ for $\alpha \in \{0.01, 0.005, 0.001\}$.

FPR / TPR denotes the true and false positive rates under the critical value $q_\alpha = 6$. The FPR refers to the probability of incorrectly identifying an unwatermarked table as containing watermark signal, while the TPR refers to the probability of correctly detecting the watermark signal in a watermarked table. Therefore, an FPR / TPR pair of $(0.00, 1.00)$ indicates an effective watermark—no false alarms and complete detection.

Table 9: Results of Monte Carlo simulation on 100 unwatermarked synthetic tables of 1K rows. Each Entry show an empirical estimation (first value) and a theoretical value (second value) under standard assumption in Lemma 3.

| Dataset | $\hat{\mu}_{\text{nwm}}/\mu_{\text{nwm}}$ | $\hat{\sigma}_{\text{nwm}}/\sigma_{\text{nwm}}$ | $\hat{q}_{0.01}/q_{0.01}$ | $\hat{q}_{0.005}/q_{0.005}$ | $\hat{q}_{0.001}/q_{0.001}$ |
|---|---|---|---|---|---|
| Adult | 0.86/1.00 | 0.62/0.71 | 2.34/2.32 | 2.59/2.57 | 3.11/3.09 |
| Magic | 2.01/2.00 | 0.87/1.00 | 2.28/2.32 | 2.52/2.57 | 3.03/3.09 |
| Shoppers | 2.03/2.00 | 0.97/1.00 | 2.51/2.32 | 2.78/2.57 | 3.34/3.09 |
| Default | 1.42/1.50 | 0.78/0.87 | 2.39/2.32 | 2.64/2.57 | 3.16/3.09 |
| Drybean | 1.77/2.00 | 0.89/1.00 | 2.52/2.32 | 2.78/2.57 | 3.35/3.09 |

For completeness, the TabularMark detector reported in our experiments is given by

$$Z = \frac{2\left(n_g - 0.5\,n_w\right)}{\sqrt{n_w}},$$

where $n_w$ is the total number of key cells and $n_g$ is the number of those cells that fall in the "green" domain.

## F.3    Implementation Details for Data Generator

**TabSyn.**    TabSyn (Zhang et al., 2024a) is an architecture designed for high-quality tabular data synthesis. It addresses the challenges of mixed-type features by mapping raw tabular inputs—including numerical, categorical, and other modalities—into a continuous latent space using a customized variational autoencoder (VAE (Kingma and Welling, 2014)). The VAE employs Transformer-based encoders and decoders to effectively model inter-column dependencies and generate token-level embeddings. In the embedding space, TabSyn leverages a score-based diffusion model with a simplified linear noise schedule, which enables efficient sampling and preserving fidelity to the original data distribution. The combination of autoencoding and latent-space diffusion allows TabSyn to generate diverse and realistic synthetic tabular data with high efficiency and quality.

**Implementation in our work.**    While our experiments are conducted using the TabSyn framework as the generative backbone, we adhere to the implementation in TabWak (Zhu et al., 2025), which is our primary baseline for comparison, to ensure fair and consistent comparison. Specifically, we replace the original score-based diffusion process in TabSyn with DDPM for training and DDIM for sampling. Therefore, our reproduced baseline results ("W/O") reported in Table 2 are closely aligned with those in Table 1 of TabWak.

The discrepancies between our "W/O" results and those reported in the original TabSyn paper (Zhang et al., 2024a) stem from the two modifications mentioned above. As reported in the TabSyn paper itself, the original score-based TabSyn model outperformed the TabSyn-DDPM variant in generation quality owing to its tailored diffusion process in continuous latent space. Additionally, the deterministic nature of the DDIM sampler may reduce data diversity compared to the original score-based SDE sampler. Nevertheless, the DDIM sampler is essential for TabWak, as its watermark detection relies on the inversion process.

In Table 10, we also present the evaluation results of our methods on synthetic tabular data generated by original TabSyn implementation. See Appendix G.1 for more empirical results and analysis.

### F.4 Implementation Details for Watermarking

**Our method.** We sample half of the column indices using a secret key to form the index set $\mathcal{I}$ in Algorithm 2. The selection of hyperparameter $\lambda$ in YJT is implemented by the Python module `scipy.stats.yeojohnson` (SciPy Developers, 2025). For the Adult, Magic and Drybean datasets, we apply Algorithm 1 to all numerical columns. For the Shoppers dataset, we apply watermarking to first 9 numerical columns. For the Default dataset, we select columns $\{0, 4, 17, 18, 19, 20, 21, 22\}$ for watermarking. We provide explanations on this implementation details in Remark 2.

**MUSE.** Following the experimental setting in Fang et al. (2025), we adopt Bernoulli as the scoring function and an adaptive column selection mechanism with three columns. We also adhere to the original configuration of $m = 2$, meaning that between two candidate rows, the one with the higher score is selected as the watermarked sample. Since TabSyn generates tabular data as a whole rather than row by row, we generate twice the target number of rows and then perform selection, consistent with the workflow illustrated in Algorithm 1 of Fang et al. (2025).

**TabWak\*.** We rigorously reproduce TabWak\* with valid bit mechanism by following all instructions provided in the official repository `https://github.com/chaoyitud/TabWak`. However, we observed a significant discrepancy between our reproduction results and those reported in the original TabWak paper (Zhu et al., 2025). Below, we clarify the source of this discrepancy.

In TabWak's detection pipeline, the suspect tabular data $\mathbf{X}$ is first mapped into a continuous latent space via the inversion of the VAE decoder to obtain an initial latent representation $\mathbf{z}_0$. This is then passed through DDIM inversion to recover the watermarked representation $\mathbf{z}_T$. In practice, TabWak codebase obtains $\mathbf{z}_0$ from $\mathbf{X}$ by solving a gradient-based optimization problem to approximate the inversion of the VAE decoder, which is formulated as:

$$\mathbf{z} = \arg\min_{\mathbf{z}} \|\mathbf{x} - f_\theta(\mathbf{z})\|_2,$$

where $f_\theta$ denotes the trained VAE decoder.

However, we found that the optimization procedure often fails to converge to the true latent code $\mathbf{z}_0$, which leads to significantly reduced detectability and robustness of the watermark signal (as reported in our paper). We also note that in the official TabWak codebase, the ground-truth $\mathbf{z}_0$ is saved during generation. Using this saved $\mathbf{z}_0$ bypasses the inversion step and yields strong

detectability, which is comparable to that reported in the original TabWak paper. But this approach is impractical in real-world detection where the ground-truth $\mathbf{z}_0$ is unavailable.

**GLW.** We set the number of "green list" intervals to $m = 5$. Since GLW was originally designed for tabular data with continuous density functions, we introduce a minor modification to extend it to mixed-type tabular data. Specifically, for entries with non-zero decimal components, we directly apply the standard method proposed in (He et al., 2024). For integer entries with non-zero units digits, we shift the decimal point one place to the left, apply the method to the transformed values, and then shift them back. To prevent significant distortion, we exclude entries with absolute values less than 1 from watermarking. GLW is applied to all numerical columns across all datasets.

**TabularMark.** Following the original experimental setup, we select the first numerical column as the watermark attribute, from which 10% of the cells are pseudorandomly chosen as key cells. The perturbation range $p$ is set to 25, and the number of unit domains $k$ to 500. To implement the matching algorithm in (Zheng et al., 2024), we extract the first five binary bits from each of five randomly selected attributes and concatenate them to form a 25-bit binary string, which serves as the primary key for each tuple.

## F.5 Implementation Details for Tabular Attacks

We implement the ten post-processing attacks for Table 3 as below:

1. **Row Del.** removes 10% of rows in a table.

2. **Col Del.** replaces 2 columns with unwatermarked values sampled from the same model.

3. **Cell Del.** replaces 10% cells with unwatermarked values sampled from the same model.

4. **G(aussian)-Noise.** adds Gaussian noise with zero mean and a standard deviation equal to 10% of each cell's value for numerical attributes.

5. **C(ategorical)-Noise.** perturbs categorical entries by randomly replacing 10% of cells with values sampled from other rows in the same column.

6. **A(daptive)-Noise.** adds Gaussian noise with zero mean and 0.1 standard deviation to standardized attributes. Specifically, we conduct the process below for each column $j \in \{0, \ldots, p-1\}$:

$$z_{ij} = \frac{x_{ij} - \mu_j}{\sigma_j}, \quad z'_{ij} = z_{ij} + \epsilon \cdot \mathcal{N}(0, 1), \quad x'_{ij} = z'_{ij} \cdot \sigma_j + \mu_j,$$

where $\epsilon = 0.1$ is the attack strength, and $\mu_j$ and $\sigma_j$ are the empirical mean and standard deviation of column $j$ in the original data. Conducting round and clip if $x'_{ij}$ is not in the valid range of column $j$ in the original data.

7. **Truncation.** truncates numerical values at the first significant digit.

8. **Quantization.** discretizes numerical columns using quantile transformation with the 10 quantile bins and maps those discrete quantile levels back to the original data domain with the inverse transform.

9. **Resample.** redistributes samples to achieve equal representation across target classes by super-sampling underrepresented classes and sub-sampling overrepresented ones.

10. **Shuffle.** randomly permutes all rows of the table.

While additive Gaussian noise (G-noise) attack adopted in our paper may destroy semantic meanings of columns with large absolute values and very small variance, we adhere to it for two main reasons. First, this ensures a fair comparison of robustness with TabWak* by replicating the evaluation setup used in its original paper (Zhu et al., 2025). Second, if our watermark signal remains highly detectable under such strong attacks that may significantly distort data utility, it is reasonable to expect stronger performance under milder perturbations. In fact, adaptive Gaussian noise (A-Noise) attack is implemented as a milder variant of Gaussian noise attack.

Additionally, although all the watermark methods including ours show great robustness to the shuffling attack, we believe that the shuffle attack should not be omitted. In practice, the existence of this cost-free row-level attack has important implications for our design. Without shuffle attack, we could use the record indices as hash seeds to deterministically sample pseudorandom bit sequences for each row, enabling accurate recovery during detection and avoiding key collision. However, the bit sequence recovery becomes vulnerable when row-ordering is no longer preserved under shuffle attack.

In Appendix G.2, we present extended robustness evaluations against above attacks with higher strength.

# G   Additional Results and Analysis

## G.1   Ablation Study

**Model-agnostic property.**   Table 10 presents the evaluation results of Tab-Drw on TabSyn implemented using the official codebase. The empirical results show the effectiveness of our method on high-fidelity synthetic data, further justifying our claim that Tab-Drw is practical and the fidelity-detectability trade-off only relies on the hyperparameters.

While our experiments are conducted using TabSyn framework as the generative backbone, we expect Tab-Drw to exhibit similar performance when applied to other synthetic tabular data generators, since Tab-Drw is a post-editing watermarking method that operates independently of the generative model's architecture or sampling procedure.

To justify this claim, we perform evaluations using two additional popular tabular data generators: TabDDPM (Kotelnikov et al., 2023) and STaSy (Kim et al., 2023). The results are presented in Table 11. Overall, our method achieves great fidelity-detectability trade-off across all three models (including TabSyn), demonstrating its effectiveness and model-agnostic property.

**YJT selection.**   In Tab-Drw, the Yeo-Johnson transformation (YJT) serves as a pre-conditioner for constructing the frequency-domain representation. By mapping each feature

Table 10: Data fidelity and watermark detectability evaluated on tables generated by original TabSyn implementation. No watermarking is denoted as "W/O". Our proposed TAB-DRW uses $(\gamma, \delta) = (0.5, 0.5)$. Fidelity metrics are averaged over 10 trials while Z-score is averaged over 100 trials.

| Datasets | Method | Fidelity Metric | | | | Z-score | |
| --- | --- | --- | --- | --- | --- | --- | --- |
| | | Density ↑ | Corr ↑ | C2ST ↑ | MLE ↑ | 1K rows ↑ | 5K rows ↑ |
| Adult | W/O | 0.993±0.001 | 0.982±0.003 | 0.994±0.001 | 0.912±0.002 | – | – |
| | **TAB-DRW** | 0.981±0.004 | 0.967±0.003 | 0.988±0.006 | 0.910±0.003 | 12.57±1.16 | 28.07±0.99 |
| Magic | W/O | 0.990±0.001 | 0.991±0.003 | 0.993±0.002 | 0.936±0.002 | – | – |
| | **TAB-DRW** | 0.983±0.003 | 0.978±0.003 | 0.991±0.004 | 0.933±0.002 | 27.11±0.77 | 61.02±0.82 |
| Shoppers | W/O | 0.985±0.003 | 0.973±0.002 | 0.964±0.003 | 0.920±0.005 | – | – |
| | **TAB-DRW** | 0.976±0.005 | 0.955±0.003 | 0.953±0.007 | 0.919±0.006 | 17.30±1.02 | 39.31±1.06 |
| Default | W/O | 0.987±0.001 | 0.952±0.001 | 0.975±0.002 | 0.764±0.004 | – | – |
| | **TAB-DRW** | 0.982±0.002 | 0.948±0.004 | 0.971±0.009 | 0.764±0.005 | 16.87±0.91 | 37.73±0.88 |
| Drybean | W/O | 0.987±0.002 | 0.992±0.003 | 0.978±0.003 | 0.911±0.006 | – | – |
| | **TAB-DRW** | 0.984±0.004 | 0.977±0.007 | 0.972±0.006 | 0.908±0.009 | 41.23±0.98 | 90.33±1.06 |

Table 11: Data fidelity and watermark detectability evaluated on tables generated by TabDDPM and STaSy. For fiedelity metrics, the first value in each entry denotes the result without watermark, while the second value denotes the result of TAB-DRW with $(\gamma, \delta) = (0.5, 0.5)$.

| Datasets | Model | Fidelity Metric | | | | Z-score | |
| --- | --- | --- | --- | --- | --- | --- | --- |
| | | Density ↑ | Corr ↑ | C2ST ↑ | MLE ↑ | 1K rows ↑ | 5K rows ↑ |
| Adult | TabDDPM | 0.982/0.967 | 0.969/0.958 | 0.975/0.973 | 0.903/0.894 | 12.44±0.96 | 29.07±1.05 |
| | STaSy | 0.887/0.883 | 0.864/0.858 | 0.408/0.423 | 0.901/0.893 | 13.07±1.22 | 29.95±1.19 |
| Magic | TabDDPM | 0.989/0.971 | 0.983/0.975 | 0.999/0.996 | 0.935/0.923 | 28.74±0.98 | 62.53±1.14 |
| | STaSy | 0.937/0.927 | 0.933/0.919 | 0.694/0.688 | 0.933/0.926 | 27.95±0.89 | 61.48±0.97 |
| Shoppers | TabDDPM | 0.972/0.959 | 0.933/0.921 | 0.834/0.832 | 0.918/0.911 | 17.94±1.24 | 39.27±1.22 |
| | STaSy | 0.906/0.898 | 0.915/0.907 | 0.548/0.553 | 0.914/0.908 | 17.49±1.06 | 38.52±1.17 |
| Default | TabDDPM | 0.985/0.982 | 0.951/0.948 | 0.971/0.967 | 0.756/0.755 | 15.27±0.94 | 35.29±0.92 |
| | STaSy | 0.942/0.940 | 0.940/0.939 | 0.681/0.677 | 0.752/0.749 | 16.65±1.00 | 37.02±1.14 |
| Drybean | TabDDPM | 0.987/0.984 | 0.971/0.960 | 0.967/0.954 | 0.892/0.894 | 40.22±1.16 | 88.59±0.94 |
| | STaSy | 0.949/0.947 | 0.919/0.912 | 0.582/0.596 | 0.890/0.891 | 39.47±1.05 | 86.98±0.91 |

toward a more Gaussian-like distribution, YJT helps to standardize heterogeneous feature scales and reduce skewness, which is essential for enabling a stable, low-distortion watermarking process in the frequency domain.

We further emphasize that YJT also improves the robustness of rank-based statistics used in our pseudorandom bit generation procedure. Specifically, transforming the feature distributions makes ranks more evenly spread and less sensitive to local density variations, which in turn improves bit consistency under post-processing perturbations.

To support this claim empirically, we include an ablation study comparing TAB-DRW with and without YJT. As shown in the Table 12, the YJT consistently yields a better trade-off between fidelity and watermark detectability, confirming its importance in our design.

**Gray code selection.** We provide additional experimental results using a 1-Gray code and compare it to our adopted variant of 2-Gray code. Specifically, we modify lines 7 and 9 in

Table 12: Ablation study on YJT. Both methods are applied to TAB-DRW with $(\gamma, \delta) = (0.5, 0.5)$. Fidelity metrics are averaged over 10 trials while Z-score is averaged over 100 trials.

| Datasets | Method | Fidelity Metric | | | | Z-score | |
| --- | --- | --- | --- | --- | --- | --- | --- |
| | | Density ↑ | Corr ↑ | C2ST ↑ | MLE ↑ | 1K rows ↑ | 5K rows ↑ |
| Adult | W/O YJT | 0.906±0.004 | 0.862±0.003 | 0.601±0.006 | 0.812±0.007 | **13.74±0.87** | **31.62±0.95** |
| | W/ YJT | **0.915±0.005** | **0.864±0.004** | **0.604±0.008** | **0.816±0.009** | 12.81±1.17 | 29.55±1.12 |
| Magic | W/O YJT | 0.907±0.004 | **0.936±0.003** | 0.666±0.002 | 0.817±0.011 | 27.17±0.95 | 60.91±0.93 |
| | W/ YJT | **0.910±0.005** | 0.935±0.003 | **0.676±0.009** | **0.818±0.014** | **27.34±0.93** | **61.42±1.02** |
| Shoppers | W/O YJT | 0.896±0.005 | 0.900±0.002 | 0.704±0.009 | 0.888±0.012 | 12.79±0.96 | 28.50±0.98 |
| | W/ YJT | **0.909±0.006** | **0.902±0.003** | **0.712±0.013** | **0.891±0.014** | **18.18±1.28** | **40.74±1.26** |
| Default | W/O YJT | 0.921±0.007 | 0.906±0.008 | 0.689±0.014 | 0.789±0.011 | 10.05±0.98 | 22.73±1.08 |
| | W/ YJT | **0.929±0.010** | **0.907±0.011** | **0.713±0.018** | **0.791±0.013** | **15.98±0.92** | **35.84±0.91** |
| Drybean | W/O YJT | 0.923±0.009 | 0.911±0.004 | 0.527±0.016 | 0.877±0.014 | 28.12±0.87 | 62.80±0.79 |
| | W/ YJT | **0.931±0.013** | **0.928±0.007** | **0.655±0.029** | **0.880±0.019** | **38.03±1.03** | **85.05±0.67** |

Table 13: Data fidelity and watermark detectability evaluated on TAB-DRW using different Gray codes for bit generation. Fidelity metrics are averaged over 10 trials while Z-scores are averaged over 100 trials.

| Datasets | Bit Gen. | Fidelity Metric | | | | Z-score | |
| --- | --- | --- | --- | --- | --- | --- | --- |
| | | Density ↑ | Corr ↑ | C2ST ↑ | MLE ↑ | 1K rows ↑ | 5K rows ↑ |
| Adult | 1-Gray code | **0.916±0.005** | 0.863±0.005 | 0.600±0.009 | **0.816±0.008** | 11.06±0.99 | 24.95±0.86 |
| | 2-Gray code | 0.915±0.005 | **0.864±0.004** | **0.604±0.008** | 0.816±0.009 | **12.81±1.17** | **29.55±1.12** |
| Magic | 1-Gray code | 0.917±0.006 | 0.936±0.003 | **0.676±0.008** | 0.818±0.014 | 21.74±0.96 | 48.78±0.97 |
| | 2-Gray code | **0.917±0.005** | **0.937±0.003** | 0.676±0.009 | **0.818±0.014** | **27.34±0.93** | **61.42±1.02** |
| Shoppers | 1-Gray code | 0.909±0.006 | **0.909±0.004** | **0.715±0.011** | **0.892±0.014** | 17.30±1.14 | 38.48±1.14 |
| | 2-Gray code | **0.909±0.006** | 0.902±0.003 | 0.712±0.013 | 0.891±0.014 | **18.18±1.28** | **40.74±1.26** |
| Default | 1-Gray code | 0.927±0.010 | **0.918±0.009** | 0.715±0.013 | **0.793±0.013** | 15.07±1.02 | 33.72±0.97 |
| | 2-Gray code | **0.929±0.010** | 0.907±0.011 | **0.717±0.018** | 0.791±0.013 | **15.98±0.92** | **35.84±0.91** |
| Drybean | 1-Gray code | **0.933±0.010** | **0.932±0.007** | **0.659±0.022** | **0.880±0.019** | 25.40±0.92 | 57.74±1.03 |
| | 2-Gray code | 0.931±0.013 | 0.928±0.007 | 0.655±0.029 | 0.880±0.019 | **38.03±1.03** | **85.05±0.67** |

Algorithm 2 as follows: We change line 7 to "**for** $j \leftarrow 1$ **to** $m$ **do**" and line 9 to "Append 1 to **S** if $k\%4 = 0$ or 3, otherwise append 0". We set the attack strengths for robustness evaluation the same as the strengthened version adopted in Appendix G.2.

From the results, we observe that using a 1-Gray code for bit generation yields a slight improvement in data fidelity, as it more closely mimics an ideal bit sampled from a Bernoulli distribution. However, its detectability and robustness decrease on several datasets, especially those dominated by continuous variables such as **Magic** and **Drybean**. We attribute this to the finer-grained rank-bin partition induced by the 1-Gray code and to the greater instability of the sum-based score for continuous features under perturbations, which makes cross-bin rank shifts more likely to happen.

**Column selection for watermarking.** In the main paper, our empirical evaluation focuses on numerical columns (including mixed continuous and discrete types) to enable a fair comparison with the other post-editing watermarking methods, GLW and TabularMark, which both suffer substantial fidelity degradation when applied to all columns. Here we also report results obtained by applying TAB-DRW to all columns, showing how different selection strategies influence the

Table 14: Robustness evaluation of TAB-DRW using different Gray codes for bit generation under the strengthened attack setting. Z-scores are averaged over 100 trials on tables with 5K rows.

| Datasets | Bit Gen. | Attacks | | | | | | | | | |
|---|---|---|---|---|---|---|---|---|---|---|---|
| | | Row Del. | Col Del. | Cell Del. | G-Noise | C-Noise | A-Noise | Truncation | Quantization | Resample | Shuffle |
| Adult | 1-Gray code | 22.30 | 10.10 | 11.19 | 12.32 | 18.11 | 21.24 | 24.95 | 16.04 | 21.49 | 24.95 |
| | 2-Gray code | **26.34** | **13.12** | **14.37** | **14.29** | **20.10** | **21.85** | **29.55** | **16.41** | **28.15** | **29.55** |
| Magic | 1-Gray code | 43.47 | 8.46 | 16.27 | 22.81 | 44.14 | 21.42 | 38.83 | 19.32 | 24.58 | 48.78 |
| | 2-Gray code | **54.85** | **11.75** | **21.60** | **33.93** | **48.38** | **29.06** | **52.62** | **26.43** | **37.61** | **61.42** |
| Shoppers | 1-Gray code | 34.36 | 13.40 | **14.00** | 35.54 | 25.05 | **13.12** | **31.00** | 23.36 | 14.34 | 38.48 |
| | 2-Gray code | **36.21** | **13.75** | 13.27 | **37.71** | **32.60** | 13.10 | 30.28 | **25.72** | **29.28** | **40.74** |
| Default | 1-Gray code | 29.84 | 19.45 | 15.05 | 21.66 | 26.14 | 12.52 | 33.72 | 11.06 | 30.08 | 33.72 |
| | 2-Gray code | **31.92** | **20.70** | **15.44** | **23.75** | **27.20** | **16.99** | **35.84** | **14.10** | **32.36** | **35.84** |
| Drybean | 1-Gray code | 51.38 | 15.48 | 16.87 | 12.73 | 48.93 | 15.50 | 22.94 | 15.31 | 39.44 | 57.74 |
| | 2-Gray code | **75.91** | **32.92** | **35.27** | **23.57** | **77.70** | **42.48** | **42.14** | **45.95** | **68.69** | **85.05** |

Table 15: Data fidelity and watermark detectability evaluated on TAB-DRW with different columns selected for watermarking. Fidelity metrics are averaged over 10 trials, and Z-scores are averaged over 100 trials.

| Datasets | Col. Selection | Fidelity Metric | | | | Z-score | |
|---|---|---|---|---|---|---|---|
| | | Density ↑ | Corr ↑ | C2ST ↑ | MLE ↑ | 1K rows ↑ | 5K rows ↑ |
| Adult | All Col. | $0.909 \pm 0.005$ | $0.859 \pm 0.005$ | $0.597 \pm 0.009$ | $0.808 \pm 0.008$ | **$14.56 \pm 0.99$** | **$32.89 \pm 1.06$** |
| | Original | **$0.915 \pm 0.005$** | **$0.864 \pm 0.004$** | **$0.604 \pm 0.008$** | **$0.816 \pm 0.009$** | $12.81 \pm 1.17$ | $29.55 \pm 1.12$ |
| Magic | All Col. | $0.914 \pm 0.006$ | $0.936 \pm 0.003$ | $0.674 \pm 0.008$ | $0.818 \pm 0.014$ | $24.66 \pm 1.08$ | $55.47 \pm 1.09$ |
| | Original | **$0.917 \pm 0.005$** | **$0.937 \pm 0.003$** | **$0.676 \pm 0.009$** | **$0.818 \pm 0.014$** | **$27.34 \pm 0.93$** | **$61.42 \pm 1.02$** |
| Shoppers | All Col. | $0.901 \pm 0.005$ | $0.897 \pm 0.003$ | $0.704 \pm 0.009$ | $0.887 \pm 0.012$ | **$19.59 \pm 1.08$** | **$43.84 \pm 1.14$** |
| | Original | **$0.909 \pm 0.006$** | **$0.902 \pm 0.003$** | **$0.712 \pm 0.013$** | **$0.891 \pm 0.014$** | $18.18 \pm 1.28$ | $40.74 \pm 1.26$ |
| Default | All Col. | $0.919 \pm 0.009$ | $0.902 \pm 0.013$ | $0.705 \pm 0.019$ | $0.787 \pm 0.011$ | **$22.21 \pm 1.03$** | **$49.96 \pm 0.99$** |
| | Original | **$0.929 \pm 0.010$** | **$0.907 \pm 0.011$** | **$0.717 \pm 0.018$** | **$0.791 \pm 0.013$** | $15.98 \pm 0.92$ | $35.84 \pm 0.91$ |
| Drybean | All Col. | $0.929 \pm 0.010$ | $0.924 \pm 0.007$ | $0.649 \pm 0.022$ | $0.878 \pm 0.019$ | **$38.35 \pm 0.89$** | **$85.47 \pm 0.73$** |
| | Original | **$0.931 \pm 0.013$** | **$0.928 \pm 0.007$** | **$0.655 \pm 0.029$** | **$0.880 \pm 0.019$** | $38.03 \pm 1.03$ | $85.05 \pm 0.67$ |

tradeoff between fidelity and detectability. In general, using more columns for watermarking improves robustness (Theorem 2 shows that the lower bound of the Z-score scales with the number of selected columns), while incurring slightly higher distortion.

The results in Table 15 show that watermarking more columns improves detectability while reducing fidelity, consistent with our theoretical analysis.

**Impact of rounding and clipping on watermark detectability.** Since the outputs of the inverse DFT and YJT are real-valued, rounding and clipping are necessary for discrete features to preserve semantic validity. However, these operations may also perturb the sign-bit alignment in the frequency domain, potentially weakening the watermark signal. Fortunately, the sign-bit alignment of TAB-DRW is highly insensitive to such mild nonlinear perturbations. In addition, because our method preserves fidelity well under appropriate choices of $(\gamma, \delta)$, clipping occurs only rarely and rounding magnitudes remain minimal.

Table 16 shows the results of an ablation study comparing Z-scores with and without rounding and clipping across five datasets, together with the frequency and magnitude of these operations. For **Magic** dataset there are no rounding or clipping happening since all the columns are continuous. For other datasets, the impact of these two post-processing operations on watermark detectability is negligible.

Table 16: Detection performance of Tab-Drw with or without the rounding and clipping operations. Z-scores are averaged over 100 trials on tables with 1K rows. "Rounding magnitude" denotes the average rounding magnitude of discrete entries, and "Clipping ratio" denotes the fraction of discrete entries that are clipped.

| Dataset | W/O round and clip | W/ round and clip | Rounding magnitude | Clipping ratio |
|---|---|---|---|---|
| Adult | $15.21 \pm 1.00$ | $12.81 \pm 1.17$ | $0.0911 \pm 0.0015$ | $0.0008 \pm 0.0004$ |
| Magic | $27.34 \pm 0.93$ | $27.34 \pm 0.93$ | $0.0000 \pm 0.0000$ | $0.0000 \pm 0.0000$ |
| Shoppers | $21.00 \pm 1.15$ | $18.18 \pm 1.28$ | $0.0969 \pm 0.0042$ | $0.0244 \pm 0.0036$ |
| Default | $17.94 \pm 0.95$ | $15.98 \pm 0.92$ | $0.0542 \pm 0.0018$ | $0.0151 \pm 0.0013$ |
| Drybean | $37.79 \pm 1.02$ | $38.03 \pm 1.03$ | $0.1285 \pm 0.0027$ | $0.0145 \pm 0.0022$ |

## G.2 Additional Robustness Evaluation

**Attacks with high strength.** In this section, we benchmark the robustness of Tab-Drw and other watermarking methods using attacks with higher strength. Specifically, we use the setting below:

1. **Row Del.** removes 20% of rows in a table.

2. **Col Del.** replaces 3 columns with unwatermarked values sampled from the same model.

3. **Cell Del.** replaces 20% cells with unwatermarked values sampled from the same model.

4. **G(aussian)-Noise.** adds Gaussian noise with zero mean and a standard deviation equal to 20% of each cell's value for numerical attributes.

5. **C(ategorical)-Noise.** perturbs categorical entries by randomly replacing 20% of cells with values sampled from other rows in the same column.

6. **A(daptive)-Noise.** adds Gaussian noise with zero mean and 0.2 standard deviation to standardized attributes.

7. **Quantization.** discretizes numerical columns using quantile transformation with the 10 quantile bins and maps those discrete quantile levels back to the original data domain with the inverse transform.

Since the **Truncation**, **Resample**, and **Shuffle** attacks are applied with fixed strength, we omit them here.

Table 17 reports the average one-sided $Z$-score over 5K rows, evaluated under the enhanced attacks. Our watermarking method still demonstrates superior robustness across all attack types and datasets, ranking either first or second. Figure 8 shows TPR@0.1%FPR versus the number of rows under three representative and strong attacks with higher strength setting. Among eight out of fifteen cases, our method reaches 1.0 TPR@0.1%FPR using only 400 rows, with the remaining seven requiring fewer than 1K rows. In contrast, baseline methods often suffer reduced true positive rates or completely lose detectability under these conditions.

To provide a more comprehensive view of robustness, we include additional empirical results under row deletion and column deletion attacks with varying deletion strengths in Figure 9. The results show that our method ranks first or second across most attack levels, demonstrating strong resilience even under high-strength attacks.

Table 17: Watermark robustness against attacks with higher strength. Average Z-score on 5K rows under seven variable-strength attacks. Each value is obtained by repeating the attacks 100 times (10 times for "TabWak*") and averaging the results. Our proposed TAB-DRW is evaluated with the hyperparameter $(\gamma, \delta) = (0.5, 0.5)$. Best performances are shown in **bold**, and second-best are underlined.

| Datasets | Method | Attacks | | | | | | |
|---|---|---|---|---|---|---|---|---|
| | | Row Del. | Col Del. | Cell Del. | G-Noise | C-Noise | A-Noise | Quantization |
| | | 20% | 3 col | 20% | 20% | 20% | 20% | 20% |
| Adult | GLW | 14.76 | 13.10 | 13.19 | 0.00 | 16.54 | 2.77 | 3.03 |
| | MUSE | 13.31 | 4.96 | 6.17 | 11.84 | 8.05 | 3.99 | 10.99 |
| | TabWak* | 14.44 | 8.05 | 7.87 | 0.02 | 15.67 | 10.23 | 5.56 |
| | TabularMark | 20.29 | **13.92** | 12.99 | 3.31 | 5.54 | 0.62 | 0.00 |
| | **TAB-DRW** | **26.34** | 13.12 | **14.37** | **14.29** | **20.10** | **21.85** | **16.41** |
| Magic | GLW | **153.98** | **123.60** | **137.64** | 0.10 | **172.20** | 0.31 | 14.08 |
| | MUSE | 31.56 | 3.70 | 9.30 | 8.39 | 33.34 | 4.06 | 0.39 |
| | TabWak* | 17.27 | 7.47 | 13.45 | 16.44 | 19.76 | 13.39 | 12.86 |
| | TabularMark | 16.09 | 10.68 | 11.74 | 0.00 | 19.39 | 0.68 | 0.00 |
| | **TAB-DRW** | 54.85 | 11.75 | 21.60 | **33.93** | 48.38 | **29.06** | **26.43** |
| Shoppers | GLW | **36.82** | **34.46** | **32.33** | 0.00 | **39.08** | 1.13 | 0.00 |
| | MUSE | 25.85 | 8.64 | 9.05 | 21.58 | 16.20 | **16.26** | 13.61 |
| | TabWak* | 8.93 | 2.26 | 0.97 | 0.00 | 10.47 | 1.22 | 0.69 |
| | TabularMark | 13.68 | 8.74 | 10.13 | 0.98 | 13.29 | 0.00 | 1.42 |
| | **TAB-DRW** | 36.21 | 13.75 | 13.27 | **37.71** | 32.60 | 13.10 | **25.72** |
| Default | GLW | 25.67 | 19.89 | **19.88** | 0.00 | 27.08 | 6.49 | 9.60 |
| | MUSE | 30.80 | 8.52 | 7.30 | 14.01 | 17.67 | 3.79 | 4.97 |
| | TabWak* | 21.96 | 12.84 | 13.49 | **23.77** | 23.70 | **18.52** | **20.25** |
| | TabularMark | 19.72 | 16.21 | 11.17 | 0.00 | 17.10 | 0.80 | 2.58 |
| | **TAB-DRW** | **31.92** | **20.70** | 15.44 | 23.75 | **27.20** | 16.99 | 14.10 |
| Drybean | GLW | **116.96** | **104.46** | **98.54** | 0.18 | **123.28** | 5.05 | 27.90 |
| | MUSE | 28.12 | 4.58 | 6.34 | 6.13 | 27.71 | 2.85 | 0.00 |
| | TabWak* | 16.01 | 0.00 | 0.00 | 11.21 | 17.53 | 10.42 | 3.43 |
| | TabularMark | 12.06 | 5.27 | 3.22 | 0.00 | 13.54 | 2.43 | 0.00 |
| | **TAB-DRW** | 75.91 | 32.92 | 35.27 | **23.57** | 77.70 | **42.48** | **45.95** |

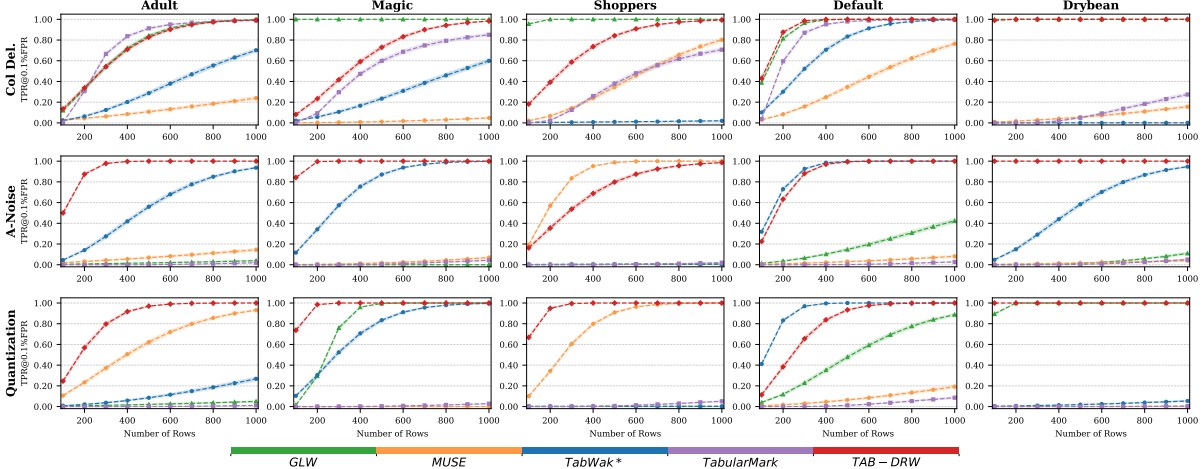

Figure 8: TPR@0.1%FPR versus row count under three representative attacks with higher strength. Dashed lines show the bootstrap mean estimate (500 resamples), and shaded regions indicate the 90% confidence interval.

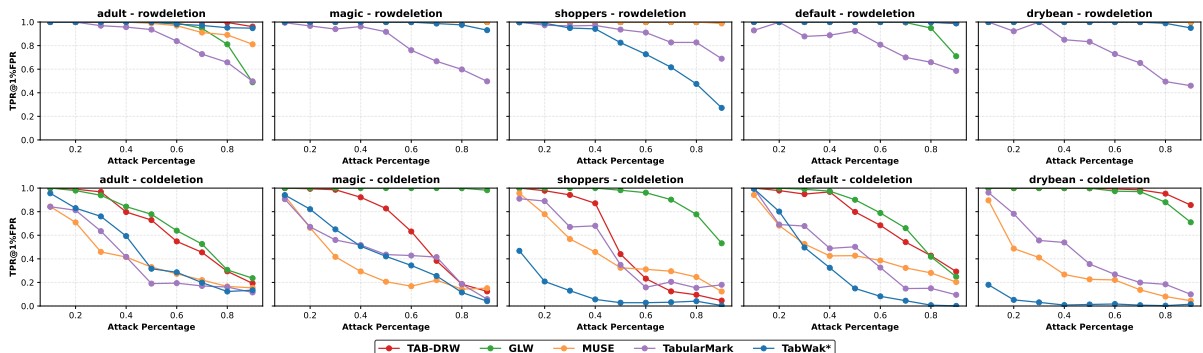

Figure 9: TPR@1%FPR versus attack strength under row and column deletion attacks. All experiments are conducted on tables with 1K rows. Each value denotes the result of 100 independent trials.

Table 18: Robustness of TAB-DRW against rewatermarking attacks of varying strength. Fidelity is averaged over four metrics across 10 independent trials, and the Z-scores are computed on tables with 5K rows and averaged over 100 independent trials. "Rewatermarking@$n$" denotes rewatermarking the table using $n$ randomly sampled keys.

| Datasets | No-attack | | Rewatermarking@1 | | Rewatermarking@3 | | Rewatermarking@10 | |
|---|---|---|---|---|---|---|---|---|
| | Fidelity | Z-score | Fidelity | Z-score | Fidelity | Z-score | Fidelity | Z-score |
| Adult | $0.799 \pm 0.006$ | $29.55 \pm 1.12$ | $0.787 \pm 0.008$ | $23.66 \pm 1.17$ | $0.772 \pm 0.006$ | $16.26 \pm 1.09$ | $0.766 \pm 0.009$ | $17.26 \pm 1.34$ |
| Magic | $0.837 \pm 0.008$ | $61.42 \pm 1.02$ | $0.822 \pm 0.008$ | $53.23 \pm 0.91$ | $0.813 \pm 0.007$ | $34.32 \pm 0.93$ | $0.799 \pm 0.008$ | $29.17 \pm 1.00$ |
| Shoppers | $0.854 \pm 0.009$ | $40.74 \pm 1.26$ | $0.847 \pm 0.008$ | $31.97 \pm 1.15$ | $0.829 \pm 0.009$ | $20.14 \pm 1.09$ | $0.813 \pm 0.008$ | $16.67 \pm 1.09$ |
| Default | $0.836 \pm 0.013$ | $35.84 \pm 0.91$ | $0.827 \pm 0.011$ | $32.85 \pm 1.00$ | $0.811 \pm 0.013$ | $19.40 \pm 1.07$ | $0.809 \pm 0.010$ | $26.28 \pm 1.18$ |
| Drybean | $0.849 \pm 0.017$ | $85.05 \pm 0.67$ | $0.832 \pm 0.017$ | $44.79 \pm 0.81$ | $0.801 \pm 0.017$ | $29.47 \pm 0.83$ | $0.806 \pm 0.014$ | $33.77 \pm 0.95$ |

Table 18 shows the fidelity degradation under rewatermarking attacks of varying rounds, serving as an extension to Table 4.

**Robustness to spoofing attacks.** We also clarify the robustness of TAB-DRW under spoofing attacks mentioned in Ngo et al. (2025). In our setting, spoofing attack refers to making unwatermarked data showcase strong watermark signal under the detection using a specific key. TAB-DRW is explicitly designed to make this extremely difficult, as detailed in Appendix B. In conclusion, the privacy-enhanced variant applies a key-dependent column permutation before the YJT and DFT, creating a large key space and yielding empirically negligible cross-key collisions (i.e., a watermark embedded with one key cannot be misdetected under another). This arises because the imaginary sign-bit alignments induced by different keys (i.e., different column orders) are approximately unrelated. Since the detection key is private, an adversary without access to it cannot efficiently tune modifications to increase the Z-score, and naive or heuristic modifications either fail to spoof the watermark or noticeably degrade data fidelity. Even if attackers imitate the imaginary sign pattern of the frequency-domain representation from a watermarked table, they still cannot produce a detectable watermark signal as long as their keys differ from the specific detection key. We refer the readers to Appendix B and Appendix G.3 for additional analysis and empirical justification.

However, even without access to the watermark key, an adversary could still attempt a model-level spoofing attack by distilling a new generator from watermarked outputs, similar to the spirit of distilling from watermarked LLMs (Gu et al., 2024). To evaluate this threat, we simulate such an attacker as follows: for each of the five datasets, we first sample a large

Table 19: Robustness of Tab-Drw against distilling-based spoofing attack. Each entry denotes the result over 100 independent trials.

| Datasets | 1K rows | | 5K rows | |
| --- | --- | --- | --- | --- |
| | Z-score | TPR@0.1%FPR | Z-score | TPR@0.1%FPR |
| Adult | 0.88±1.03 | 0.01 | 1.62±1.04 | 0.05 |
| Magic | 1.87±1.12 | 0.15 | 3.75±1.08 | 0.74 |
| Shoppers | 1.21±0.86 | 0.01 | 2.19±0.97 | 0.13 |
| Default | 0.62±0.94 | 0.00 | 1.28±1.06 | 0.03 |
| Drybean | 2.02±0.98 | 0.12 | 4.13±1.25 | 0.79 |

corpus of synthetic tables from a TabSyn model equipped with Tab-Drw, and then train a fresh TabSyn model only on these watermarked samples, using the same architecture and training protocol as the original generator. We then generate tables from the distilled model and test them with our standard detector.

The results in Table 19 indicate that this distilled generator generally fails to successfully spoof the Tab-Drw watermark. For **Adult**, **Shoppers**, and **Default**, tables with 1K rows produced by the distilled model are statistically indistinguishable from unwatermarked data. In contrast, for **Magic** and **Drybean**, we observe a moderate watermark signal. Empirically, datasets dominated by continuous attributes appear more vulnerable to this kind of spoofing by distillation, which we hypothesize is due to the stronger and more smoothly distributed watermark signal embedded in their continuous feature distributions.

## G.3 Privacy-Enhanced TAB-DRW Evaluation

Table 20: Data fidelity and watermark detectability of privacy-enhanced TAB-DRW under varying watermark keys. All experiments use $(\gamma, \delta) = (0.5, 0.5)$. Fidelity metrics are averaged over 10 trials, and the $Z$-score is averaged over 100 trials.

| Datasets | Key | Fidelity Metric | | | | Z-score | |
| | | Density ↑ | Corr ↑ | C2ST ↑ | MLE ↑ | 1K rows ↑ | 5K rows ↑ |
|---|---|---|---|---|---|---|---|
| Adult | W/O | 0.922±0.001 | 0.872±0.001 | 0.611±0.004 | 0.824±0.005 | – | – |
| | Key 48 | 0.912±0.003 | 0.862±0.003 | 0.598±0.008 | 0.814±0.009 | 11.98±0.97 | 26.48±1.07 |
| | Key 496 | **0.916±0.003** | **0.869±0.004** | 0.601±0.006 | **0.819±0.009** | **16.69±1.24** | **37.39±1.37** |
| | Key 928 | 0.915±0.005 | 0.864±0.004 | **0.604±0.008** | 0.816±0.009 | 12.81±1.17 | 29.55±1.12 |
| Magic | W/O | 0.917±0.001 | 0.945±0.003 | 0.672±0.004 | 0.823±0.007 | – | – |
| | Key 48 | **0.915±0.004** | 0.934±0.005 | **0.676±0.009** | **0.821±0.009** | 24.06±0.75 | 53.19±0.86 |
| | Key 496 | 0.915±0.004 | **0.939±0.005** | 0.666±0.007 | 0.819±0.012 | 27.33±1.04 | 61.17±1.08 |
| | Key 928 | 0.910±0.005 | 0.935±0.003 | 0.676±0.009 | 0.818±0.014 | **27.34±0.93** | **61.42±1.02** |
| Shoppers | W/O | 0.919±0.002 | 0.910±0.001 | 0.704±0.005 | 0.902±0.012 | – | – |
| | Key 48 | **0.912±0.004** | **0.907±0.002** | 0.706±0.008 | **0.893±0.015** | 16.15±1.11 | 36.11±1.16 |
| | Key 496 | 0.904±0.005 | 0.902±0.003 | 0.698±0.011 | 0.889±0.015 | 14.84±1.10 | 34.28±1.16 |
| | Key 928 | 0.909±0.006 | 0.902±0.003 | **0.712±0.013** | 0.891±0.014 | **18.18±1.28** | **40.74±1.26** |
| Default | W/O | 0.930±0.001 | 0.907±0.001 | 0.717±0.003 | 0.797±0.009 | – | – |
| | Key 48 | **0.929±0.010** | 0.906±0.007 | **0.715±0.010** | **0.791±0.013** | 13.56±1.02 | 30.32±0.98 |
| | Key 496 | 0.929±0.010 | **0.907±0.010** | 0.714±0.012 | 0.791±0.013 | 13.82±0.96 | 30.90±0.99 |
| | Key 928 | 0.929±0.010 | 0.907±0.011 | 0.713±0.018 | 0.791±0.013 | **15.98±0.92** | **35.84±0.91** |
| Drybean | W/O | 0.932±0.001 | 0.935±0.001 | 0.640±0.003 | 0.878±0.009 | – | – |
| | Key 48 | 0.930±0.007 | 0.926±0.005 | 0.649±0.014 | **0.881±0.017** | 37.21±0.84 | 83.14±0.91 |
| | Key 496 | 0.930±0.007 | **0.929±0.006** | 0.631±0.025 | 0.875±0.011 | 30.98±0.91 | 70.27±0.83 |
| | Key 928 | **0.931±0.013** | 0.928±0.007 | **0.655±0.029** | 0.880±0.019 | **38.03±1.03** | **85.05±0.67** |

**Data fidelity vs. watermark detectability.** Table 20 shows that privacy-enhanced TAB-DRW achieves consistently high data fidelity and detectability across three randomly sampled keys. Although minor variations exist, they remain within an acceptable range, indicating that users need not devote much effort to tuning the key. Additionally, the empirical results further strengthen our claim that the key-dependent variability in the frequency-domain representation does not substantially affect watermark distortion or detectability.

**Multi-key scenarios.** Under a deployment scenario with multiple watermark key holders, we evaluate potential key collision, i.e., how many different keys $\kappa$ in Algorithm 3 & 2 can be used for a dataset without leading to false positives during detection. In practice, there exists an upper bound on the number of watermark keys that can be supported without introducing elevated false positives. And this capacity is influenced by the number of dataset columns. The cross-key confusion matrices in Table 21 present empirical results on the ability to detect and distinguish between different watermark keys, demonstrating the superiority of our method in avoiding potential key collisions in multi-user scenarios.

Table 21: Multi-key evaluation on five benchmarks. The randomly selected keys along the horizontal axis are used for sampling, while those along the vertical axis are used for detection: FPR/TPR(diagonal) of 1K independent trials under threshold $q_\alpha = 6$ on 1K rows.

| Dataset | Detection key | Sampling keys | | | | |
| --- | --- | --- | --- | --- | --- | --- |
| | | Key 48 | Key 275 | Key 496 | Key 643 | Key 928 |
| Adult | Key 48 | 1.000 | 0.000 | 0.000 | 0.000 | 0.000 |
| | Key 275 | 0.000 | 0.998 | 0.000 | 0.000 | 0.000 |
| | Key 496 | 0.000 | 0.000 | 1.000 | 0.001 | 0.000 |
| | Key 643 | 0.000 | 0.007 | 0.000 | 0.996 | 0.000 |
| | Key 928 | 0.000 | 0.000 | 0.000 | 0.000 | 1.000 |
| Magic | Key 48 | 1.000 | 0.000 | 0.000 | 0.000 | 0.000 |
| | Key 275 | 0.000 | 1.000 | 0.000 | 0.000 | 0.000 |
| | Key 496 | 0.000 | 0.000 | 1.000 | 0.001 | 0.000 |
| | Key 643 | 0.000 | 0.000 | 0.000 | 1.000 | 0.000 |
| | Key 928 | 0.000 | 0.000 | 0.000 | 0.000 | 1.000 |
| Shoppers | Key 48 | 1.000 | 0.000 | 0.000 | 0.000 | 0.000 |
| | Key 275 | 0.000 | 1.000 | 0.000 | 0.000 | 0.000 |
| | Key 496 | 0.000 | 0.000 | 1.000 | 0.000 | 0.000 |
| | Key 643 | 0.000 | 0.000 | 0.000 | 1.000 | 0.000 |
| | Key 928 | 0.000 | 0.000 | 0.000 | 0.000 | 1.000 |
| Default | Key 48 | 1.000 | 0.000 | 0.000 | 0.000 | 0.000 |
| | Key 275 | 0.002 | 1.000 | 0.000 | 0.000 | 0.000 |
| | Key 496 | 0.000 | 0.000 | 1.000 | 0.000 | 0.000 |
| | Key 643 | 0.000 | 0.000 | 0.000 | 1.000 | 0.000 |
| | Key 928 | 0.000 | 0.000 | 0.000 | 0.000 | 1.000 |
| Drybean | Key 48 | 1.000 | 0.000 | 0.000 | 0.000 | 0.000 |
| | Key 275 | 0.000 | 1.000 | 0.000 | 0.000 | 0.000 |
| | Key 496 | 0.000 | 0.000 | 1.000 | 0.000 | 0.000 |
| | Key 643 | 0.000 | 0.000 | 0.000 | 1.000 | 0.004 |
| | Key 928 | 0.000 | 0.000 | 0.000 | 0.000 | 1.000 |

# H    Potential Limitations

Although TAB-DRW demonstrates strong robustness under a broad range of post-processing and adaptive attacks, it is not intended to be unconditionally robust against arbitrarily strong transformations. First, sufficiently aggressive attacks that heavily disturb the rank statistic used in Algorithm 2, or repeated rewatermarking over many rounds, can eventually reduce watermark detectability, although such attacks also cause noticeable fidelity degradation. Second, the statistical power of the detector depends on the number of effective DFT entries and the number of available rows; therefore, tables with very few usable columns or very small sample size are inherently less favorable for reliable detection. Third, our additional spoofing experiments suggest that datasets dominated by continuous attributes may be more vulnerable to distillation-based spoofing than mixed-type datasets. Finally, in multi-key deployment, the number of mutually distinguishable keys is finite and depends on the table dimension, so the practical capacity of the

key space is dataset-dependent.

# I  The Use of Large Language Models (LLMs)

We acknowledge the use of a large language model (LLM) solely for polishing writing. The LLM was not employed for developing mathematical theorems, proofs, or any part of the experimental results or analysis. All text edited with the assistance of the LLM has been carefully reviewed to ensure that it does not introduce plagiarism or scientific misconduct. We take full responsibility for all content presented in this work.

