# OpenReview forum: "Robust Spectral Watermark for Synthetic Tabular Data"
_SLADS/Section_B — Accepted by SLADS_Section_B_

### Review · Reviewer_fmM3 · 2026-03-13

**Summary Of Contributions:**

The paper studies watermarking for tabular data and proposes a method called Tab-Drw. The method first applies YJT and normalization to columns, and then performs DFT along rows. Due to properties of the DFT, the number of effective columns is roughly half of the original column size. The method then generates bits using a tree-based procedure with a secret key, and modifies values according to the resulting pseudorandom numbers. Applying inverse DFT, normalization, and YJT produces the final watermarked table. The paper provides testing statistics based on Z-scores, as well as theoretical analysis of column-wise differences and robustness under Gaussian/sub-Gaussian sampling settings. Empirical evaluations on five benchmark datasets from multiple perspectives are also presented.

**Audience:**

Yes

**Broader Impact Concerns:**

No major ethical concerns are identified in this work.

**Claims And Evidence:**

Yes

**Requested Changes:**

The main suggested improvements are described in the **Major** comments. Addressing them would improve the readability and completeness of the paper. However, these issues are **not critical** to my recommendation. The paper would still be acceptable in my view even if these points are only partially addressed.

**Strengths And Weaknesses:**

## Strengths

- The proposed method is clearly presented and can be applied to tabular data without relying on neural network models. The use of a rank-based mechanism further improves robustness.
- The paper provides theoretical analysis on column-wise differences and robustness under Gaussian and sub-Gaussian attacks.
- The simulation results demonstrate good performance of the proposed method.
- Ablation experiments under different hyperparameter settings are well organized.



## Weaknesses

### Major

- Algorithm 2 would benefit from a more detailed explanation. Providing a concrete example with step-by-step illustrations could help readers better understand the procedure. Such an example could be included in the appendix.
- It is unclear how the random numbers $\zeta_{i,j}$ are generated from the pseudorandom bits. More details on this generation process would improve clarity and reproducibility.
- Section 4.3 demonstrates the performance of the watermarking method under 10 attacks. It would be helpful to further explain why the proposed method remains robust under these attacks. For instance, the robustness against "three representative and strong attacks" can be elaborated.
- Are there potential attacks or scenarios under which the proposed method may fail or become sensitive? A discussion of possible limitations would strengthen the paper.

### Minor

- Page 6, Line -6 in **Refinement: Soft variant**, "When $\gamma=0$ and $\delta=0$": it seems that "or" may be more appropriate. From my understanding, no watermark would be embedded if either condition is satisfied. A similar issue appears on Page 11, Line -5.

---

> ### Author Response · Authors · 2026-03-16
> **Response to Reviewer fmM3**
>
> # Response to Reviewer fmM3
>
> We thank the reviewer for the constructive feedback and the positive assessment of the clarity of the method, the theoretical analysis, and the empirical performance. Below we address the comments point by point.
>
> ---
>
> ### **Q1: Algorithm 2 would benefit from a more detailed explanation and a concrete step-by-step example.**
>
> Thank you for this suggestion. In the revised version, we have added a concrete example in Appendix C that provides a step-by-step illustration of Algorithm 2.
>
> ---
>
> ### **Q2: It is unclear how the random numbers $\zeta_{i,j}$ are generated from the pseudorandom bits.**
>
> We appreciate the opportunity to clarify this point. As stated in the paper (page 6, line 10; page 8, line 8), the symbols $\zeta_t$ and $\zeta_{i,j}$ denote the pseudorandom bits themselves, so there is no additional step in which other random numbers are generated from these bits.
>
> ---
>
> ### **Q3: Section 4.3 should better explain why the method remains robust under the 10 attacks, especially under the three representative strong attacks.**
>
> The main intuition behind the strong robustness, as is discussed in the second paragraph of Section 2.3 and further elaborated in Section 3.2, is that TAB-DRW does not rely on modifying any single cell. Instead, it distributes the watermark across rows and effective frequency entries, while detection aggregates sign-alignment statistics over the entire table. This leads to several forms of robustness.
>
> First, deletion-type attacks remove only part of the table, so a substantial portion of aligned frequency signs remains available for detection. Second, noise and discretization attacks primarily perturb amplitudes, whereas the detector relies only on the sign alignment of the imaginary parts. Third, row-wise bit recovery is based on a key-selected rank statistic, which is relatively stable under moderate perturbations; moreover, neighboring rank bins differ by only one bit pair, so even rank shifts typically introduce only limited mismatches. Fourth, row shuffling does not harm detection because the method does not depend on a fixed row order. Finally, the use of a secret key makes adversarial attacks significantly more difficult.
>
> ---
>
> ### **Q4: Are there attacks or scenarios under which the proposed method may fail or become sensitive?**
>
> While the current experiments and appendix already suggest several limitation regimes, we agree that these should be discussed more explicitly. First, sufficiently aggressive attacks that heavily disrupt row ranks, or many rounds of rewatermarking, can eventually reduce detectability, although such transformations also lead to noticeable fidelity degradation. Second, tables with very few usable columns or very few rows naturally provide fewer effective DFT entries and therefore lower statistical power. Third, the appendix shows that datasets dominated by continuous attributes can be somewhat more vulnerable to distillation-based spoofing than mixed-type datasets. Fourth, in multi-key deployment, the number of safely distinguishable keys is finite and depends on the table dimension. We have added Appendix H to further discuss these potential limitations.
>
> ---
>
> ### **Q5: Minor wording issue in the soft variant.**
>
> We thank the reviewer for pointing this out and have revised the manuscript accordingly.

---

> > ### Comment · Reviewer_fmM3 · 2026-03-17
> >
> > The authors have adequately addressed my concerns. No more comments.

---

### Review · Reviewer_K2Tb · 2026-03-24

**Summary Of Contributions:**

The paper proposes a post-editing watermarking scheme for synthetic tabular data with mixed feature types. The main methodology applies the Discrete Fourier Transform to embed the watermark signal in the frequency domain by altering the imaginary components. Building on this, the paper presents an algorithm for pseudorandom bit generation utilized in the alterations and a novel hypothesis test for watermark detection. The theoretical section examines the impact on distributional properties and derives a lower bound on the expected value of the $z$-score under Gaussian-noise attacks, providing justifications for the fidelity and robustness of the proposed method. Extensive evaluations on real datasets explore the synthetic data fidelity, watermark detectability, and robustness against attacks in comparison to existing methods.

**Audience:**

Yes

**Broader Impact Concerns:**

The authors acknowledge the broader societal harms and risks of adopting AI-generated tabular data into fields such as healthcare, finance, and public policy. An efficient and robust watermarking scheme can help ensure accountability for generated datasets. These ideas are already present in the paper and abstract, so a Broader Impact Statement may not be required.

**Claims And Evidence:**

Yes

**Requested Changes:**

## Changes that could strengthen the work
* I think the paper would benefit from clarifying how the $z$-score is computed for all baselines. While equation (3) defines the $z$-score based on the alignment count with pseudorandom bits, it is also noted that for TabularMark, the $z$-score is replaced by another formula from Zheng et al., 2024. However, the computation for the remaining methods remains unclear to me.
* To further validate the computational efficiency of the proposed method in Section 4.5, it may be worth discussing the relative prevalence of embedding vs detection in watermarking applications. In particular, if detection is invoked more often in practice, it would further underscore the advantage of TAB-DRW over TabWak* and TabularMark, which have higher detection costs but smaller embedding costs.

## Questions
* Table 2 shows that TAB-DRW ranks first or second in most of the datasets for the C2ST metric. Do the authors have any insights into this possible specialization in data-fidelity performance? Specifically, is there a reason the proposed method could be particularly well-suited for a certain data fidelity metric?

## Minor Changes
* In the proof of Theorem 2, page 36, both of the vectors in the variance equation should be $s_t$?
* In the proof of Theorem 3, page 39, 'following’ is missing an ‘o’

**Strengths And Weaknesses:**

* The paper is well written and structured.
* The paper has good visualizations that aid in understanding the main ideas.
* The paper clearly addresses limitations in existing watermarking schemes for synthetic tabular data.
* The paper presents a diligent robustness analysis. From a theoretical standpoint, the lower bound of the z-score expected value for tables under Gaussian-noise attacks provides justification for watermark detectability for tables with iid sub-Gaussian rows. The empirical stress testing further demonstrates robustness against post-processing attacks such as deletion, noise addition, discretization, and structural changes. Figure 5 demonstrates how statistical power changes as attacks intensify.

---

> ### Author Response · Authors · 2026-03-25
> **Response to Reviewer K2Tb**
>
> # Response to Reviewer K2Tb
>
> We thank the reviewer for the constructive feedback and for the positive assessment of the writing, visualizations of ideas, and robustness analysis. Below we address the comments point by point.
>
> ---
>
> ### **Q1: It would help to clarify how the Z-score / detection statistic is computed for all baselines.**
>
> We thank the reviewer for pointing this out and have clarified it more clearly in Appendix F.2.
>
> More specifically, **TAB-DRW, GLW, MUSE, and TabWak\*** all use the **same one-sided Z-score formula (equation 3)** as the table-level detection statistic: we first compute a row-level detection statistic $T_i$ for each method, estimate its null mean and standard deviation from unwatermarked tables, and then standardize the table-level average into a one-sided Z-score. What differs across methods is the underlying row-level statistic $T_i$:
>
> - for **TAB-DRW**, it is the count of frequency entries whose imaginary signs align with the recovered pseudorandom bits.
> - for **GLW**, it is the count of entries falling into the key-selected green intervals.
> - for **MUSE**, it is the row-level Bernoulli score induced by its percentile-based keying rule.
> - for **TabWak\***, it is the valid-bit accuracy computed from the reversed latent noise after DDIM inversion.
>
> The only exception is **TabularMark**. Since TabularMark watermarks a single numerical attribute and uses only 10% of its cells as key cells, we follow the detector of Zheng et al. (2024), which is defined in terms of key-cell counts in the green domain.
>
> ---
>
> ### **Q2: Section 4.5 would benefit from discussing the relative prevalence of embedding versus detection in practice.**
>
> We appreciate this suggestion and believe it will strengthen the practical motivation for the runtime comparison. First, watermark embedding is typically a **one-time operation** performed when a synthetic table is generated. Relative to the underlying table generation time itself—about **~2 seconds** at the scale of Table 5 in our runs—the embedding overhead is almost negligible for **GLW**, **TabWak\***, **TabularMark**, and **TAB-DRW**. The exception is **MUSE**, whose embedding is more expensive because it generates multiple candidates and then performs a selection step.
>
> Second, watermark detection may be invoked **repeatedly** in practice, for example by auditors or platform operators. This effect is even more pronounced in **multi-key deployment scenarios**, where the same suspect table may need to be checked against multiple candidate keys. In this setting, detection efficiency becomes more important. This further strengthens the practical advantage of **TAB-DRW**, especially relative to **TabWak\***, whose detector must invoke the inverse diffusion process. We have revised Section 4.5 to make this claim more explicit.
>
> ---
>
> ### **Q3: Why does TAB-DRW often rank first or second on the C2ST metric?**
>
> We thank the reviewer for this insightful question. We agree that different watermarking methods may favor different fidelity metrics, as these metrics capture different aspects of distributional preservation. In particular, C2ST is classifier-based and therefore more sensitive to whether watermarking alters the **joint structure** of the data in a way that makes watermarked samples easier to distinguish from the original ones.
>
> We believe the relatively strong C2ST performance of TAB-DRW is mainly due to its **joint frequency-domain perturbation mechanism**. Rather than applying entry-wise modifications in the original space, TAB-DRW first maps each row, after transformation and standardization, into a frequency-domain representation where each coefficient depends jointly on all features. As a result, modifying selected spectral coefficients leads to a coordinated perturbation across the entire row after inverse transformation.
>
> This design is well aligned with C2ST. Since C2ST evaluates distinguishability through a learned classifier, preserving the global geometry and dependency structure of the data is more important than minimizing local entry-wise distortion alone. Compared with localized perturbations in the original feature space, the joint spectral perturbation in TAB-DRW is more likely to maintain this multivariate structure. Therefore, although TAB-DRW is not explicitly designed for C2ST, its mechanism is naturally better suited to fidelity metrics that emphasize **multivariate distinguishability**.
>
> ---
>
> ### **Q4: Minor corrections.**
>
> We thank the reviewer for catching these typos and have revised the manuscript accordingly.

---

> > ### Comment · Reviewer_K2Tb · 2026-03-26
> >
> > Thank you for the clarifications, those were my only comments.

---

### Decision · Action_Editor_bd4R · 2026-04-02

**Recommendation:** Accept as is

**Audience:**

The paper addresses a timely and relevant problem at the intersection of statistics and AI, and would be of clear interest to the SLADS Section B audience.

**Claims And Evidence:**

This paper proposes an efficient post-editing watermarking scheme for synthetic tabular data. The idea is novel, and the methodology is both technically sound and well-developed. The authors provide solid theoretical support alongside comprehensive experimental evaluations, which together offer convincing evidence for the effectiveness of the proposed approach.